# c-FLIP is crucial for IL-7/IL-15-dependent NKp46+ ILC development and protection from intestinal inflammation in mice

Ute Bank[1,13], Katrin Deiser[1,13], Carlos Plaza-Sirvent[1,2], Lisa Osbelt[3,4], Amelie Witte[1], Laura Knop[1], Rebecca Labrenz [1], Robert Jänsch[1], Felix Richter[1], Aindrila Biswas[5], Ana C. Zenclussen[6], Eric Vivier [7,8,9], Chiara Romagnani [10,11], Anja A. Kühl[12], Ildiko R. Dunay[5], Till Strowig [4], Ingo Schmitz [1,2,14] & Thomas Schüler [1,14 ✉]

NKp46+ innate lymphoid cells (ILC) modulate tissue homeostasis and anti-microbial immune responses. ILC development and function are regulated by cytokines such as Interleukin (IL)−7 and IL-15. However, the ILC-intrinsic pathways translating cytokine signals into developmental programs are largely unknown. Here we show that the anti-apoptotic molecule cellular FLICE-like inhibitory protein (c-FLIP) is crucial for the generation of IL-7/IL-15-dependent NKp46+ ILC1, including conventional natural killer (cNK) cells, and ILC3. Cytokine-induced phosphorylation of signal transducer and activator of transcription 5 (STAT5) precedes up-regulation of c-FLIP, which protects developing NKp46+ ILC from TNF-induced apoptosis. NKp46+ ILC-specific inactivation of c-FLIP leads to the loss of all IL-7/IL-15-dependent NKp46+ ILC, thereby inducing early-onset chronic colitis and subsequently microbial dysbiosis; meanwhile, the depletion of cNK, but not NKp46+ ILC1/3, aggravates experimental colitis. In summary, our data demonstrate a non-redundant function of c-FLIP for the generation of NKp46+ ILC, which protect T/B lymphocyte-sufficient mice from intestinal inflammation.

[1] Institute of Molecular and Clinical Immunology, Medical Faculty, Otto-von-Guericke University, Magdeburg, Germany. [2] Systems-Oriented Immunology and Inflammation Research Group, Department of Immune Control, Helmholtz Centre for Infection Research, Braunschweig, Germany. [3] Institute of Medical Microbiology and Hospital Hygiene, Medical Faculty, Otto-von-Guericke University, Magdeburg, Germany. [4] Microbial Immune Regulation Research Group, Helmholtz Centre for Infection Research, Braunschweig, Germany. [5] Institute of Inflammation and Neurodegeneration, Medical Faculty, Otto-von-Guericke University, Magdeburg, Germany. [6] Experimental Obstetrics and Gynecology, Medical Faculty, Otto-von-Guericke University, Magdeburg, Germany. [7] Centre d'Immunologie de Marseille-Luminy, Aix Marseille Université, Inserm, CNRS, Marseille, France. [8] Service d'Immunologie, Hôpital de la Timone, Assistance Publique-Hôpitaux de Marseille, Marseille, France. [9] Innate Pharma Research Labs., Innate Pharma, Marseille, France. [10] Innate Immunity, German Rheumatism Research Center (DRFZ), Leibniz Association, Berlin, Germany. [11] Medical Department I, Charité - University Medical Center Berlin, Berlin, Germany. [12] Charité – Universitätsmedizin Berlin, corporate member of Freie Universität Berlin, Humboldt Universität zu Berlin, and Berlin Institute of Health, iPATH, Berlin, Germany. [13] These authors contributed equally: Ute Bank, Katrin Deiser. [14] These authors jointly supervised this work: Ingo Schmitz, Thomas Schüler. ✉email: thomas.schueler@med.ovgu.de

nnate lymphoid cells (ILC) modulate immune responses and tissue homeostasis at multiple levels. Reminiscent to the classification of helper T cell subsets, different types of ILC can be distinguished based on their cytokine and transcription factor expression profiles[1]. Type 1 NKp46+T-bet+ (ILC1) produce Interferon-γ (IFN-γ) and contribute e.g. to *Toxoplasma gondii* clearance[2]. Conventional NK cells (cNK) represent a sub-population of ILC1, which is characterized by the co-expression of T-bet and Eomes[1]. Distinct from T-bet+Eomes− helper-like ILC1, cNK synthesize cytotoxic molecules, such as perforins and granzymes as well as death ligands (DLs) like tumor necrosis factor α (TNF), TNF-related apoptosis inducing ligand (TRAIL) and FAS ligand (FASL) enabling them to induce target cell apoptosis[3]. On the contrary, NKp46−GATA3+ ILC2 produce Interleukin (IL)−5 and IL-13 and promote helminth rejection[4], while NKp46+/−RORγt+ ILC3 support anti-bacterial responses in mice lacking T and B lymphocytes[5]. Furthermore, ILC3 regulate tissue homeostasis and regeneration, e.g., in the intestine and lung[6]. For example, ILC3-derived IL-22 promotes intestinal epithelial stem cell regeneration[7], providing an explanation for the tissue-protective effect of IL-22 in the course of dextran sodium sulfate (DSS)-induced colitis[8].

Most studies defining immune modulatory and tissue protective functions of NKp46+ ILC3 were performed with T and B lymphocyte (T/B)-deficient mice. However, recent evidence suggests that T lymphocytes can compensate for the lack of NKp46+ ILC3[9–11]. Whether the degree of compensation is determined by the experimental system or whether ILC3 are generally of limited importance in a T/B-competent host is still unclear.

Developing and mature ILC express a multitude of cytokine receptors including those for IL-15 and IL-7[12]. Both cytokines signal via signal transducer and activator of transcription 5 (STAT5)[13] and are crucial for ILC development and survival[2,14]. For example, IL-15 withdrawal activates the intrinsic apoptosis pathway in cNK[15]. This is associated with the failure to repress expression of the pro-apoptotic molecule Bcl2-interacting mediator of cell death (Bim), the loss of anti-apoptotic myeloid cell leukemia-1 (Mcl-1) and the subsequent death of cNK[15]. IL-7 and IL-15 are produced by e.g., stromal fibroblasts[16,17] and intestinal epithelial cells (IEC)[6,18], which create the cytokine environment required for the maintenance of ILC homeostasis and function[12]. However, ILC-intrinsic signaling molecules required for the conversion of environmental cues into developmental programs are largely unknown. In order to identify such signaling molecules and to define unique functions of intestinal NKp46+ ILC, we made use of NKp46iCre mice[19]. These mice are T/B-competent and allow targeted gene inactivation in NKp46+ ILC.

Here, we show that the anti-apoptotic molecule cellular FLICE-like inhibitory protein (c-FLIP), a master regulator of the extrinsic apoptosis pathway[20], is a target of STAT5-dependent cytokine signaling in NKp46+ ILC. The NKp46-specific inactivation of c-FLIP leads to the loss of NKp46+ ILC1, including cNK, and ILC3 without affecting T and B lymphocyte homeostasis. Furthermore, we provide evidence that cytokine-dependent c-FLIP induction prevents a TNF-induced suicide program in developing NKp46+ ILC. Finally, we define a non-redundant function of c-FLIP-dependent NKp46+ cNK in the intestine. Specifically, different from mice only lacking NKp46+ ILC3, mice lacking NKp46+ cNK fail to recover from acute colitis and develop early signs of chronic disease. Furthermore, although not affecting the establishment of a normal commensal microbiota in the steady state, NKp46+ ILC counteract inflammation-associated commensal dysbiosis. In summary, we show that the anti-apoptotic molecule c-FLIP is crucial for the cytokine-dependent development of NKp46+ cNK/ILC1 and ILC3 and subsequent protection from intestinal inflammation by cNK, a function that cannot be compensated by other immune cells.

## Results

**c-FLIP is essential for the development of NKp46+ ILC.** c-FLIP is a master regulator of the extrinsic apoptosis pathway[20] and protects immune and non-immune cells from caspase 8-dependent apoptosis[21,22]. The expression of c-FLIP is crucial for T cell development[23] and is up-regulated by activated human and mouse T cells[24]. IL-15 is crucial for the development of T-bet+ ILC1 including cNK[25,26]. Whether and how the external apoptosis pathway is involved in IL-15-dependent ILC development and function was unclear. In order to identify a functional link between IL-15 signaling and *Cflar* (encoding c-FLIP) gene regulation, CD3−NK1.1+NKp46+ ILC were purified from spleens of C57BL/6 (B6) mice and cultured for 16 h in the presence or absence of IL-15. The murine *Cflar* gene encodes for two isoforms of c-FLIP, the long form c-FLIP$_L$ and the short Raji isoform c-FLIP$_R$[27]. Quantitative real-time PCR (RT-qPCR) revealed that NKp46+ ILC significantly up-regulate the mRNA of the long isoform c-FLIP$_L$, but not c-FLIP$_R$, in response to IL-15 (Fig. 1a). Elevated *Cflar* gene activity correlated with the phosphorylation of STAT5 and elevated c-FLIP protein levels (Fig. 1b). The blockade of STAT5 phosphorylation by pimozide led to impaired IL-15-dependent c-FLIP up-regulation (Fig. 1b). Hence, c-FLIP is a target of STAT5-dependent IL-15 signaling in mature NKp46+ ILC.

Next, we investigated whether c-FLIP is required for the development of IL-15/STAT5-dependent NKp46+ ILC. For this purpose, conditional STAT5 (STAT5fl/fl) and c-FLIP (c-FLIPfl/fl) knockout mice were crossed to NKp46iCre-transgenic mice to generate NK$^{\Delta STAT5}$ and NK$^{\Delta c\text{-}FLIP}$ mice, respectively. NKp46iCre-transgenic mice harboring wildtype alleles of the respective target gene served as NK$^{WT}$ controls. Similar to IL-15-deficient (IL-15−/−) and IL-15 receptor α-deficient (IL-15R−/−) mice, numbers of splenic NKp46+ ILC were strongly reduced in NK$^{\Delta STAT5}$ and NK$^{\Delta c\text{-}FLIP}$ mice (Fig. 1c). IL-15 deprivation causes cNK apoptosis[28]. Accordingly, residual splenic IL-15−/− and IL-15R−/− NKp46+ ILC showed increased rates of caspase-3/7 activity (Fig. 1d). The frequencies of NKp46+ ILC expressing active caspase-3/7 were also elevated in NK$^{\Delta STAT5}$ and NK$^{\Delta c\text{-}FLIP}$ mice (Fig. 1d).

Furthermore, we observed increased frequencies of immature double negative (DN) NK1.1−NKp46− and single positive (SP) NK1.1+NKp46− cells in conjunction with a reduced abundance of double positive (DP) NK1.1+NKp46+ ILC in spleens and bone marrow (BM) of IL-15−/−, IL-15R−/−, NK$^{\Delta STAT5}$, and NK$^{\Delta c\text{-}FLIP}$ mice (Fig. 1e, f). Of note, the development of other immune cells remained unaltered in NK$^{\Delta c\text{-}FLIP}$ mice (Supplementary Fig. 1). Altogether, our results demonstrate that c-FLIP is crucial for the IL-15/STAT5-dependent development of NKp46+ ILC in the BM and their survival in the periphery.

**c-FLIP protects ILC precursors from TNF-induced apoptosis.** ILC1 produce effector molecules such as IFN-γ and DLs like TNF and TRAIL[29]. In the absence of c-FLIP, human cNK become sensitive to their own effector molecules and undergo apoptosis[30]. We therefore hypothesized that the loss of NKp46+ ILC in NK$^{\Delta c\text{-}FLIP}$ mice resulted from death receptor (DR)-induced apoptosis during ILC1 development. To elucidate whether effector genes are activated in BM ILC, we first analyzed IFN-γ reporter mice expressing eYFP under control of the *Ifng* promoter. As shown in Fig. 2a, *Ifng* promoter activity increased from the NK1.1−NKp46− DN to the NK1.1+NKp46− SP stage and reached its maximum at the NK1.1+NKp46+ DP stage. Similarly, TNF mRNA levels increased progressively from the DN to the DP stage (Fig. 2b). Of note, DR mRNAs for TRAIL-R2, FAS and TNF-RI were also detectable throughout ILC development, the latter being particularly high in DP cells (Fig. 2c). In addition, c-FLIP$_L$ and c-FLIP$_R$ mRNAs were expressed at all developmental

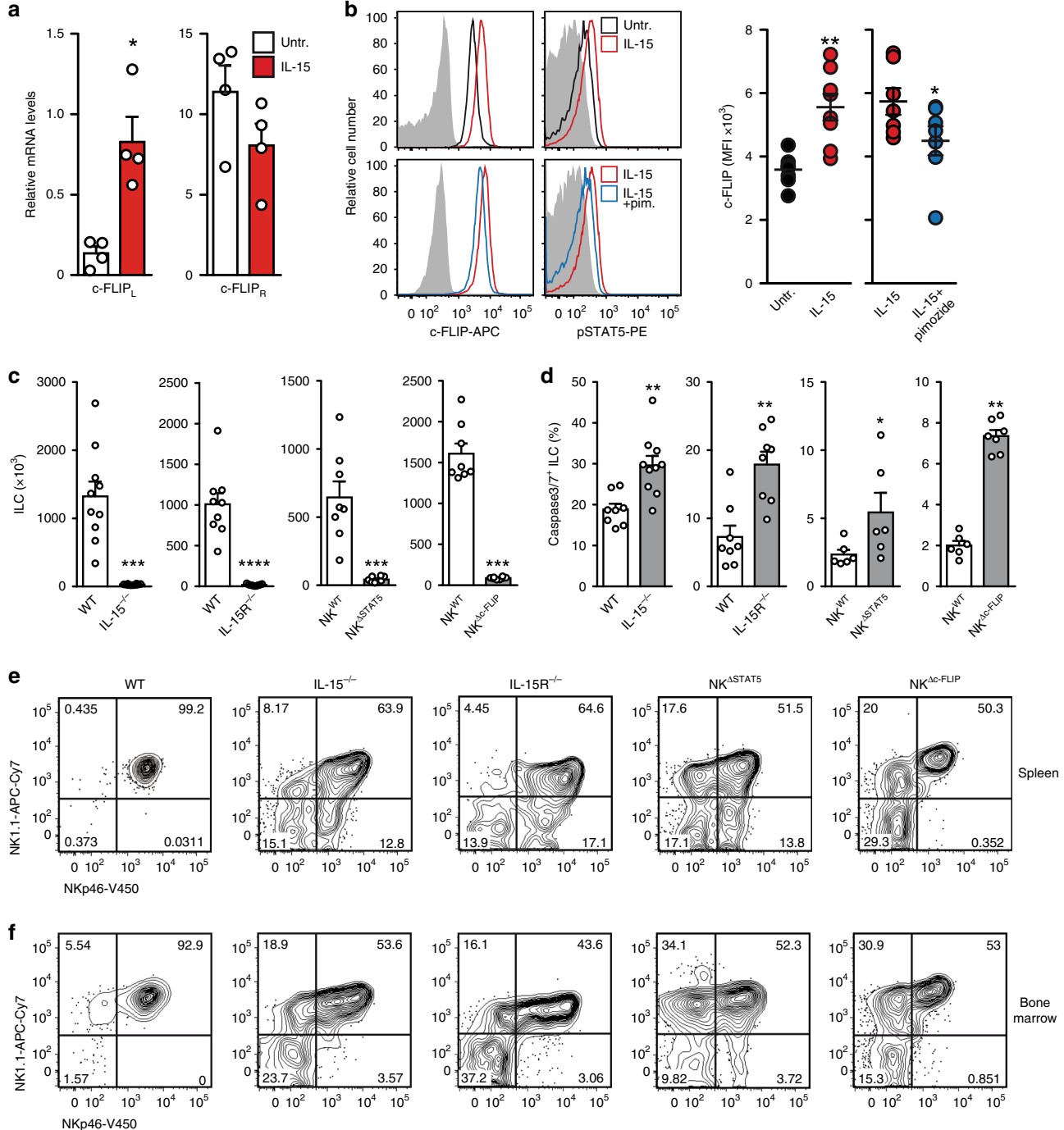

**Fig. 1 c-FLIP is crucial for IL-15-dependent ILC development. a** NK1.1$^+$NKp46$^+$ ILC were FACS-purified from spleens of C57BL/6 (B6) wild type (WT) mice and incubated for 16 h w/wo IL-15. c-FLIP$_L$ and c-FLIP$_R$ mRNA levels were quantified by RT-qPCR. Data are from four independent experiments. **b** B6 splenocytes were cultured for 16 h in the presence (red) or absence of IL-15 (black; upper row). Alternatively, cells were cultured for 16 h in the presence of IL-15 with (blue) or without the STAT5 inhibitor pimozide (red; bottom row). Relative levels of c-FLIP and phosphorylated STAT5 (pSTAT5) were determined by flow cytometry. Shown are representative histograms (fluorescence minus one controls in gray) after gating on Lin$^-$CD122$^+$NK1.1 $^+$NKp46$^+$ ILC. Graphs show mean fluorescence intensities (MFI) from 2–3 independent experiments with a total of 7–8 mice per group. **c–e** Cells from spleen and **f** BM of the indicated mouse lines were analyzed by flow cytometry. **c–f** After gating on Lin$^-$CD122$^+$ cells, **c** absolute cell numbers ($n = 8$–10) and **d** frequencies of active caspase3/7$^+$ apoptotic NK1.1$^+$NKp46$^+$ ILC were determined ($n = 6$–10). **e, f** Shown are representative contour plots from at least two independent experiments with a total of **e** 8–10 or **f** 6–10 mice per group. Numbers indicate percentages. **a–d** Data represent mean $+/\pm$ SEM. Source data are provided as source data file; *$p \leq 0.05$; **$p \leq 0.005$; ***$p \leq 0.001$; ****$p \leq 0.0001$. **a, c, d** Two-tailed Mann–Whitney $U$ test; **b** two-tailed Wilcoxon matched-pairs signed rank test.

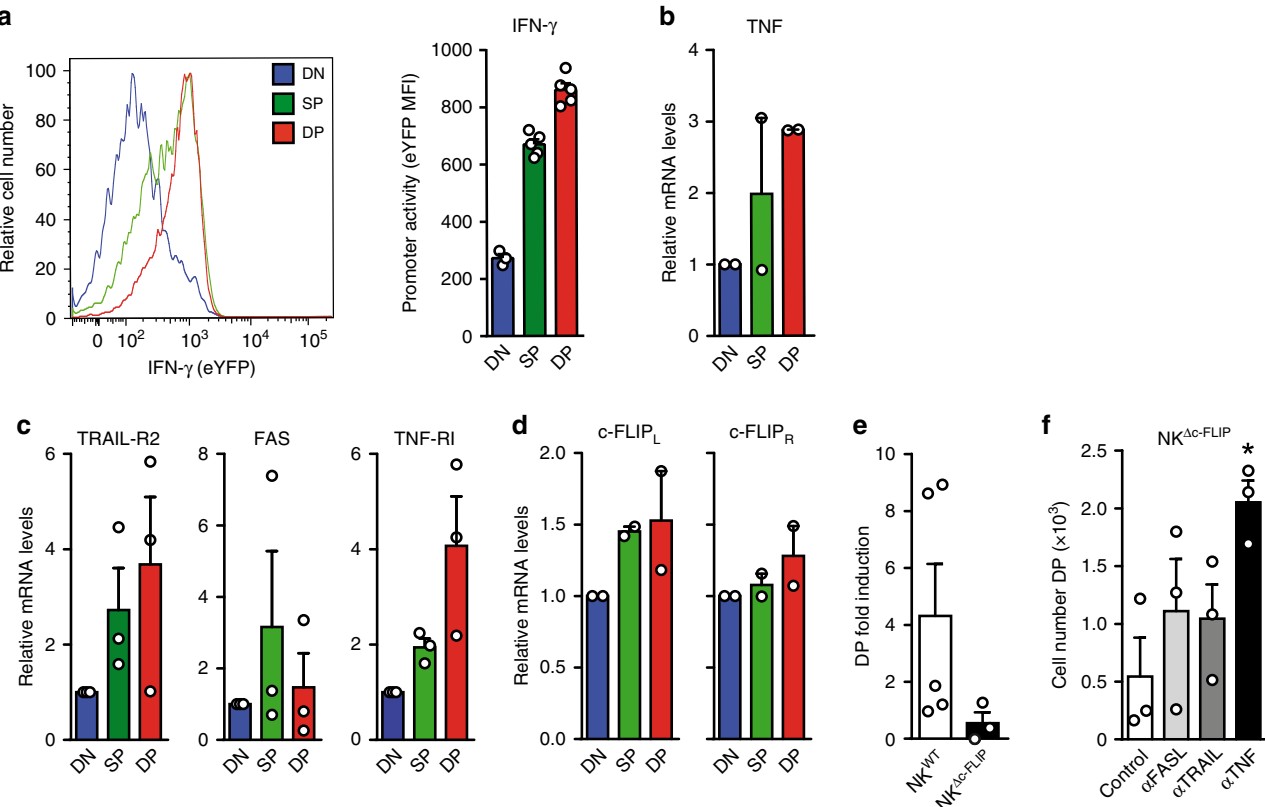

**Fig. 2 The generation of NKp46+ ILC requires c-FLIP-mediated protection from TNF-induced apoptosis. a** To determine *Ifng* promoter activity in unstimulated ILC precursors, freshly isolated BM cells from eYFP-transgenic IFN-γ reporter mice were analyzed by flow cytometry. Mean fluorescence intensities (MFI) were determined for eYFP after gating on Lin−CD122+ DN, SP, and DP cells. Representative histograms and pooled results (bar diagrams, mean + SEM) from two independent experiments with a total of five mice are shown (DN = 3 data points). **b–d** Lin−CD122+ DN, SP, and DP cells were FACS-purified from BM of B6 mice and the indicated mRNAs were quantified by RT-qPCR. mRNA levels of DN cells were set to 1 and all other values were calculated in relation. Shown are pooled results (mean + SEM) from **b**, **d** two and **c** three independent experiments. **e**, **f** Lin−CD122+ DN cells from **e** NK^WT and NK^Δc-FLIP mice or **f** NK^Δc-FLIP mice alone were enriched by FACS-sorting. **f** To block DR-function, αFASL, αTRAIL, or αTNF antibodies were added to the indicated cultures. **e**, **f** After 9 days of culture, numbers of viable DP cells were calculated in relation to those at day 0. **f** Absolute cell numbers are shown. Data (mean + SEM) were pooled from 3–5 independent experiments. **a–f** Source data are provided as source data file; **e**, **f** *p ≤ 0.05; **p ≤ 0.005; ***p ≤ 0.001; ****p ≤ 0.0001 (paired Student's *t* test).

stages (Fig. 2d). Interestingly, only c-FLIP_L mRNA showed a maturation-associated increase (Fig. 2d) similar to TNF (Fig. 2b), TRAIL-R2 and TNF-RI (Fig. 2c).

In order to elucidate the impact of DLs on c-FLIP-dependent ILC development, we established an in vitro culture system. DN cells were enriched from BM of NK^WT and NK^Δc-FLIP mice and cultured for 9 days. In accordance with our in vivo data, the generation of DP cells was impaired in the absence of c-FLIP (Fig. 2e). Importantly, this effect could be recovered most effectively by the antibody-mediated blockade of TNF (Fig. 2f).

Together, our results demonstrate that TNF and TNF-RI up-regulation are programmed events in the course of NKp46+ ILC development. The partial recovery of NK^Δc-FLIP ILC development by αTNF treatment strongly suggests that the cytokine-induced, STAT5-dependent up-regulation of c-FLIP_L (Fig. 1a, b) is crucial for protection of developing ILC from TNF-induced apoptosis.

**IL-7- and IL-15-dependent intestinal NKp46+ ILC require c-FLIP.** The small intestine (SI) harbors NKp46+T-bet+ ILC1, including T-bet+Eomes+ cNK, as well as NKp46−GATA3+ ILC2 and NKp46+/−CD4+/−RORγt+ ILC3[12]. In order to analyze whether c-FLIP affects intestinal ILC homeostasis, leukocytes were isolated from the small intestinal lamina propria of NK^WT and NK^Δc-FLIP mice. The abundance of NKp46+NK1.1+ ILC

(Fig. 3a, e) including T-bet+ ILC1 (Fig. 3b, e), NK1.1+Eomes+ cNK (Fig. 3c, e) and NKp46+RORγt+ ILC3 (Fig. 3d, e) were strongly reduced in the SI of NK^Δc-FLIP mice. Importantly, numbers of NKp46−CD4+/−RORγt+ ILC3 (Fig. 3d, f) and NKp46−GATA3+ ILC2 (Fig. 3f) remained unaltered in NK^Δc-FLIP mice. Hence, NK^Δc-FLIP mice lack NKp46+ ILC but have normal numbers of NKp46− ILC in the SI. Similar data were obtained with NK^ΔSTAT5 mice (Fig. 3g, h) further emphasizing the functional link between STAT5 and c-FLIP.

The development of cNK/ILC1 largely relies on IL-15[2,25,31] (Supplementary Fig. 2) rather than IL-7 signaling. The latter point is exemplified by the fact that cNK numbers in spleen and SI did not differ significantly between RAG1−/− and RAG1−/− × IL-7R−/− mice (Fig. 3i, j). On the contrary, numbers of all other SI NKp46+ and NKp46− ILC subsets were strongly reduced in RAG1−/− × IL-7R−/− mice (Fig. 3j, k). IL-15 can partially compensate for the lack of IL-7R signaling[32] providing an explanation for the survival of some residual ILC in RAG1−/− × IL-7R−/− mice (Fig. 3j, k). Of note, this compensatory effect was less efficient for NKp46+RORγt+ ILC3, which were more severely affected by the lack of IL-7R signaling than NKp46−RORγt+ ILC3 (Fig. 3l; lower panel).

IL-7R and IL-15R signaling converge at the level of STAT5 phosphorylation which precedes c-FLIP up-regulation (Fig. 1b).

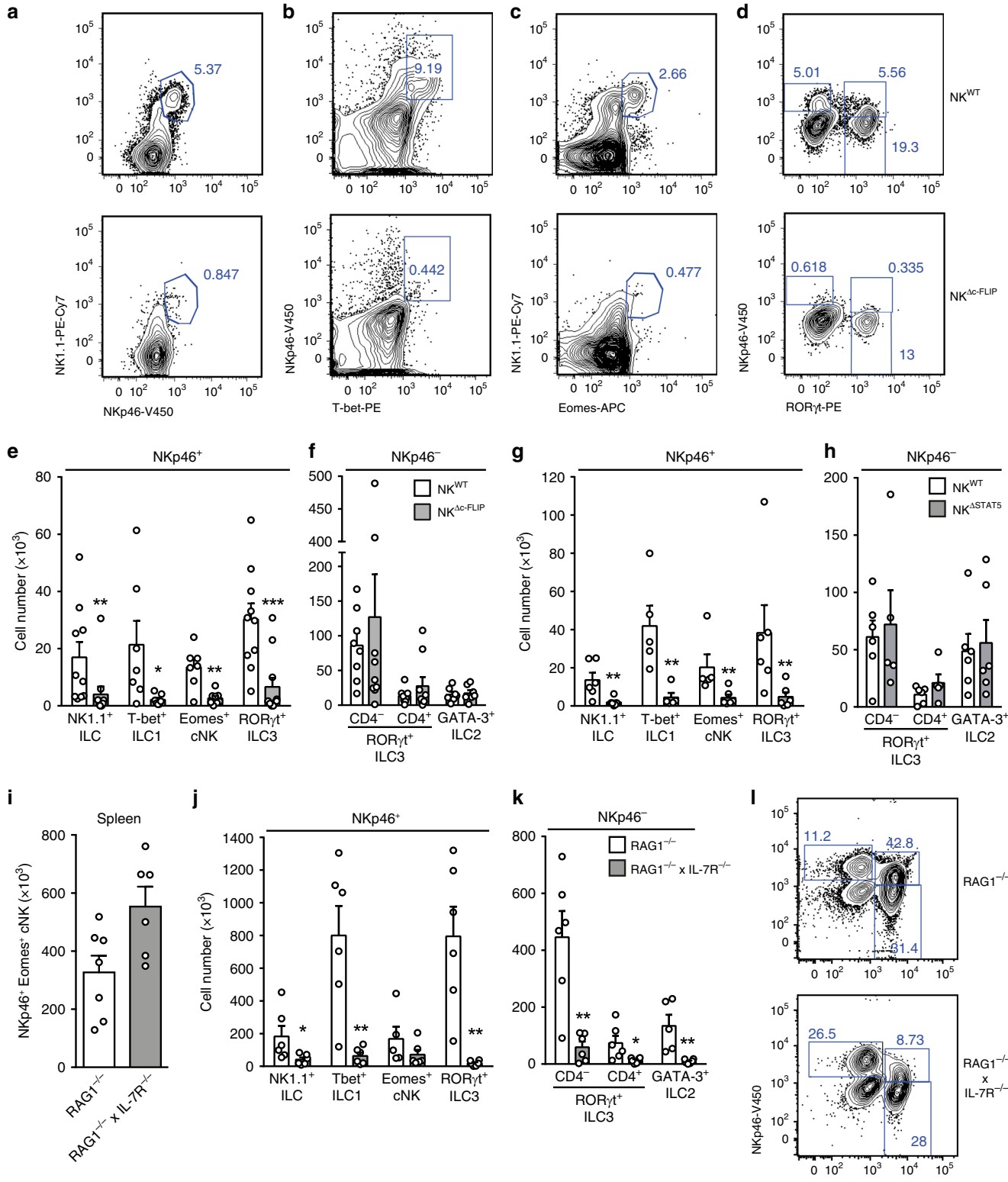

**Fig. 3 NK$^{\Delta c\text{-FLIP}}$ mice lack IL-7- and IL-15-dependent NKp46$^+$ ILC in the small intestine. a–h** and **j–l** Leukocytes were isolated from the small intestinal (SI) lamina propria of **a–f** NK$^{\Delta c\text{-FLIP}}$, **g**, **h** NK$^{\Delta STAT5}$, **j–l** RAG1$^{-/-}$ × IL-7R$^{-/-}$ and respective control mice. **i** Spleen cells from RAG1$^{-/-}$ × IL-7R$^{-/-}$ and RAG1$^{-/-}$ mice were analyzed. **a–h** and **j–l** To discriminate between different subsets of NKp46$^+$ (ILC: NK1.1$^+$; ILC1: T-bet$^+$, cNK: Eomes$^+$; ILC3: ROR$\gamma$t$^+$) and NKp46$^-$ ILC (ILC3: ROR$\gamma$t$^+$; ILC2: GATA3$^+$), cells were analyzed by flow cytometry after gating on Lin$^-$CD45$^+$ cells. **e–k** Absolute cell numbers are shown as mean + SEM. **a–d** and **l** Shown are representative contour plots and **e–k** pooled results from 3 to 5 independent experiments with a total of **e**, **f** 10–11, **g**, **h** 6, **i** 6–7 and **j**, **k** 6 mice per group. **e** For NKp46$^+$ ILC, ILC1, cNK, and ILC3, 10-11, 7, 7–8, and 10-11 samples were analyzed, respectively. **f** For NKp46$^-$ CD4$^-$ ILC3, CD4$^+$ ILC3 and ILC2, 8-9, 8-9 and 7 samples were analyzed, respectively. **g**, **h** 5-6, **i** 6-7, **j**, **k** 5-6 samples were analyzed for the indicated ILC subsets. **a–d** and **l** Numbers indicate percentages. **e–k** Source data are provided as source data file; *$p \leq 0.05$; **$p \leq 0.005$; ***$p \leq 0.001$; ****$p \leq 0.0001$ (two-tailed Mann–Whitney $U$ test).

Hence, the simultaneous lack of IL-15- and IL-7-dependent NKp46[+] ILC1 and ILC3 in NK[Δc-FLIP] and NK[ΔSTAT5] mice strongly suggests that both cytokines promote ILC development and survival via the STAT5-dependent induction of c-FLIP.

**c-FLIP-dependent NKp46[+] ILC protect mice from acute colitis.** Inflammatory bowel disease (IBD) is associated with pronounced changes in ILC1 and ILC3 frequencies and functions[29]. Whether and how the absence of NKp46[+] ILC1 and ILC3 affects the course of IBD was unclear. In order to address this subject, NK[Δc-FLIP] and control mice were treated with DSS for 5 days. The administration of DSS causes IEC damage, loss of epithelial integrity, innate immune cell activation, and results in IBD-like symptoms[33]. To monitor disease progression, relative body weight (Fig. 4a), feces consistency (Fig. 4b), and the degree of intestinal bleeding (Fig. 4c) were determined and the overall colitis score was calculated (Fig. 4d). As shown in Fig. 4a–d, disease severity was strongly increased in NK[Δc-FLIP] mice. Accordingly, the specific colon weight was strongly increased in NK[Δc-FLIP] mice indicating edema formation and increased cellularity (Fig. 4e). Furthermore, the colon length was significantly decreased in NK[Δc-FLIP] mice (Fig. 4f, g) and inflammation-related alterations in tissue architecture were more pronounced (Fig. 4h). When colonic immune cell infiltrates were analyzed, we observed the almost complete absence of NKp46[+]NK1.1[+] ILC including T-bet[+] ILC1, Eomes[+] cNK and RORγt[+] ILC3 (Fig. 4i, j) in NK[Δc-FLIP] mice. On the contrary, numbers of NKp46[−] ILC did not differ significantly between NK[Δc-FLIP] and NK[WT] mice (Fig. 4k). Importantly, CD45[+]CD11b[+] granulocytes, in particular pro-inflammatory Ly6C[+]Ly6G[+] neutrophils, rather than Ly6C[−]Ly6G[+/−] monocytes, accumulated more efficiently in the inflamed colon of NK[Δc-FLIP] mice (Fig. 4l, m). Thus, the lack of c-FLIP-dependent, NKp46[+] ILC promotes acute colitis and cannot be compensated by T or B lymphocytes.

**NKp46[+] ILC protect mice from early onset of chronic colitis.** Repeated DSS administration causes chronic colitis[33]. Since it was unclear whether this is also affected by NKp46[+] ILC, NK[WT] and NK[Δc-FLIP] mice were treated repetitively with DSS. In agreement with the results from the acute colitis model (Fig. 4a–d), disease severity was significantly more pronounced in NK[Δc-FLIP] than in NK[WT] mice after the first 5-day period of DSS treatment (Fig. 5a). After termination of the first DSS treatment, NK[WT] mice had recovered completely until day 15. This was not the case for NK[Δc-FLIP] mice. They failed to recover completely until day 19, when the second 5-day period of DSS administration was initiated (Fig. 5a). Importantly, NK[Δc-FLIP] mice rapidly developed maximum disease scores comparable to those following the first phase of DSS treatment (Fig. 5a). NK[WT] mice showed elevated disease scores at day 25, which were above those observed at the peak of the first treatment cycle but still significantly below those of NK[Δc-FLIP] mice (Fig. 5a).

Following the second DSS cycle, NK[WT] mice did not recover completely demonstrating the establishment of chronic colitis. Still, recovery was significantly less efficient in NK[Δc-FLIP] mice (Fig. 5a). After an additional phase of DSS treatment, disease scores in NK[WT] mice were slightly above those observed in the previous phase but still below those in NK[Δc-FLIP] mice (Fig. 5a). Analysis of colon samples at day 42 revealed a significant decrease in colon length (Fig. 5b) and an increase in specific colon weight in NK[Δc-FLIP] mice (Fig. 5c). Furthermore, inflammation-related histopathological changes were more pronounced in the colon of NK[Δc-FLIP] mice (Fig. 5d–f) correlating well with their increased disease severity (Fig. 5a).

The cytokine milieu has a major impact on the course of IBD[34]. We therefore determined cytokine production in supernatants of cultured colon samples isolated at days 3 and 10 after the initiation of DSS treatment. Except reduced levels of IFN-γ and GM-CSF in NK[Δc-FLIP] mice, levels of IL-22, IL-10, IL-12p70, TNF, IL-17A, IFN-β, IL-6 and MCP-1 did not differ significantly between mouse strains at day 3 (Fig. 5g; upper row). At day 10, IL-12p70, MCP-1, and GM-CSF levels were significantly lower in NK[Δc-FLIP] supernatants. On the contrary, all other cytokines were produced at comparable amounts (Fig. 5g; lower row). It is important to stress out that GM-CSF was the only cytokine that was reduced in NK[Δc-FLIP] samples at both time points. Given that the relative importance of a particular cytokine correlates positively with the persistence of its production[35], we hypothesized that GM-CSF is of particular importance for disease control in our model. In order to test whether the number of GM-CSF-producing ILC is altered in the inflamed colon of NK[Δc-FLIP] mice, colonic lamina propria leukocytes (LPL) were analyzed by flow cytometry at day 5 of DSS treatment. As shown in Fig. 5h, numbers of GM-CSF[+]NKp46[+], but not NKp46[−], ILC were significantly reduced in NK[Δc-FLIP] mice. GM-CSF promotes epithelial cell recovery after DSS treatment and GM-CSF deficiency is associated with disease aggravation[36]. Hence, increased disease severity and impaired regeneration in NK[Δc-FLIP] mice (Fig. 5a–f) correlated with (i) the reduced production of GM-CSF in colon samples and (ii) lower numbers of GM-CSF[+]NKp46[+] ILC in the colon of NK[Δc-FLIP] mice (Fig. 5g, h). In summary, our data (Fig. 5a–h) provide evidence for a function of c-FLIP-dependent NKp46[+] ILC1/3 in T/B-sufficient mice. These cells are crucial to limit intestinal inflammation, facilitate recovery and thereby prevent the early onset of chronic intestinal inflammation.

**NKp46[+] ILC counter inflammation-related commensal dysbiosis.** Intestinal homeostasis is maintained by a multitude of interactions between immune cells, IEC and the commensal microbiota[6]. Disruption of this complex network is frequently associated with aberrant immune responses, subsequent inflammation and microbial dysbiosis[37,38]. Particularly, we have shown that the severity of DSS colitis is influenced by specific interactions between microbial communities and host immune pathways with distinct changes in microbiota composition being sufficient for exacerbation of the disease[39]. We therefore asked next whether the presence or absence of NKp46[+] ILC affects the relative abundance of pro- and/or anti-inflammatory bacteria in NK[WT] and NK[Δc-FLIP] mice. For this purpose, feces samples were collected before induction of chronic DSS colitis (before DSS) and at the end of the observation period (after DSS). To minimize potential cage-specific variations in microbiota composition, NK[WT] and NK[Δc-FLIP] mice littermates were, whenever possible, co-housed permanently. Feces samples were analyzed using 16S rRNA gene sequencing of the V4 region[40] and microbiota composition was compared between NK[WT] and NK[Δc-FLIP] mice (Fig. 6a–c). Our analyses revealed a complex pattern of community structures in NK[WT] and NK[Δc-FLIP] mice. The relative contribution of factors including "Genotype", "Cage", and "Time point" to variability within the microbiota demonstrated time point-dependent differences between both mouse lines (Fig. 6a; $R^2 = 0.214$, $p < 0.001$) indicating that repeated colitis induction had a lasting impact on the microbiome composition. Notably, genotype-dependent differences in microbiome composition were not visible before induction of DSS colitis (Fig. 6a, b; ADONIS values). Thus, NKp46[+] ILC do not have a major impact on the composition of the commensal microbiota under steady-state conditions. Under inflammatory conditions, however, the presence or absence of NKp46[+] ILC in NK[WT] or NK[Δc-FLIP] mice,

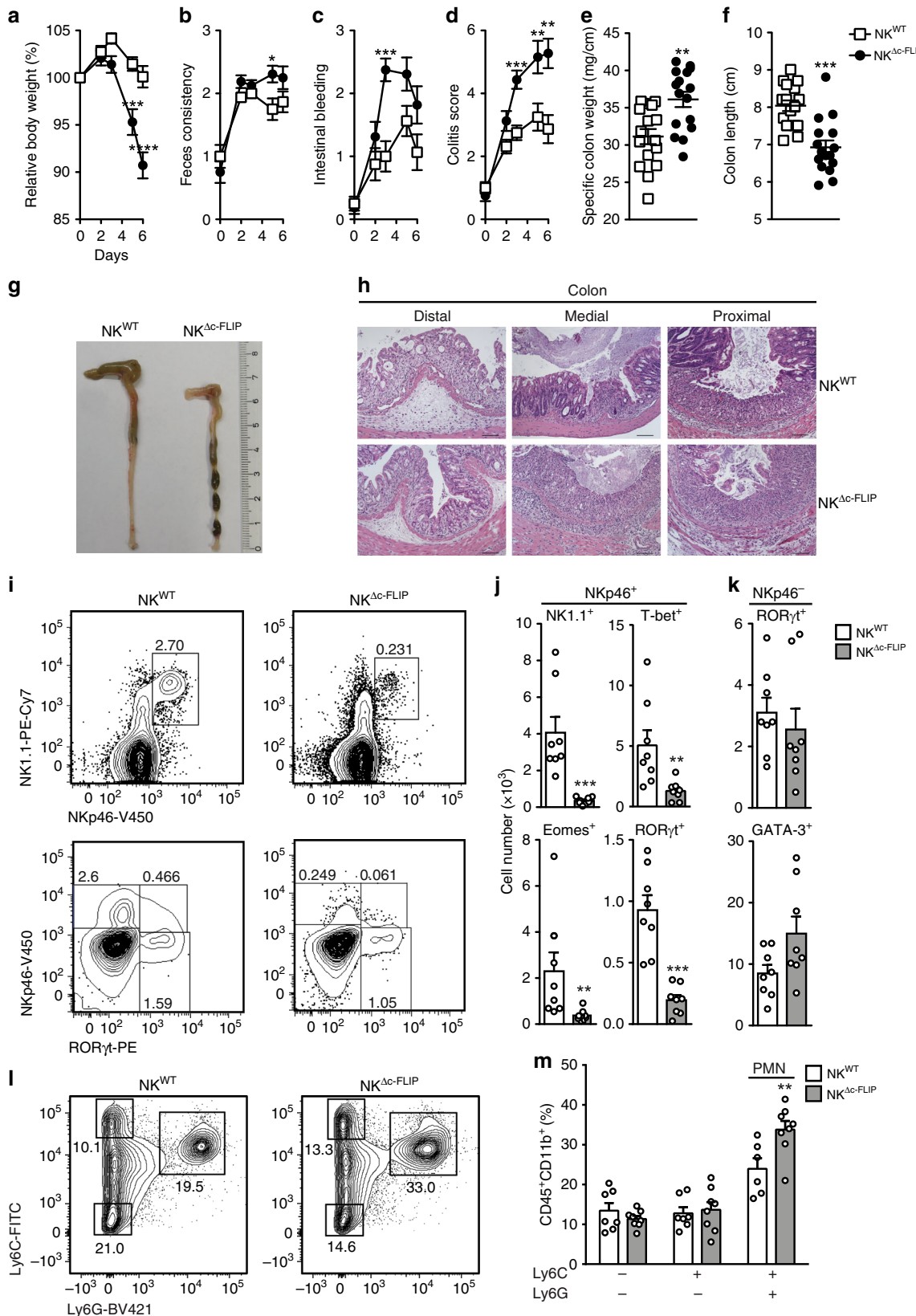

respectively, is associated with genotype-specific changes in the microbiota (Fig. 6c; $R^2 = 0.123$, $p < 0.01$).

Additionally, we compared the bacterial taxa in the commensal microbiota of $NK^{WT}$ and $NK^{\Delta c\text{-}FLIP}$ mice. In line with Fig. 6a–c, the composition of the commensal microbiota was very similar

prior to the induction of colitis (Fig. 6d). In contrast, chronic DSS colitis induced overt changes in microbiome composition in both mouse strains and distinct changes between them (Fig. 6e).

We used the linear discriminant analysis (LDA) effect size (LEfSe) method considering only operational taxonomic units

**Fig. 4 c-FLIP-dependent NKp46$^+$ ILC protect mice from acute intestinal inflammation. a–m** NK$^{WT}$ and NK$^{\Delta c\text{-}FLIP}$ mice were treated with DSS for 5 days. **a** Relative body weight, **b** feces consistency and **c** the degree of intestinal bleeding were measured on a daily basis to calculate **d** the overall colitis score. At day 7, **e** the specific weight and **f, g** the length of the colon were determined. **g, h** Shown are representative photographs of **g** colon and cecum (scale unit = cm) as well as **h** H&E–stained tissue sections from NK$^{WT}$ and NK$^{\Delta c\text{-}FLIP}$ mice ($n = 6$/group) at day 7 (scale bars = 100 μm). **i–m** At day 7, colonic lamina propria leukocytes were analyzed by flow cytometry. **i–k** After gating on Lin$^-$CD45$^+$ cells, the different subsets of NKp46$^+$ (ILC: NK1.1$^+$; ILC1: T-bet$^+$, cNK: Eomes$^+$; ILC3: RORγt$^+$) and NKp46$^-$ ILC (ILC3: RORγt$^+$; ILC2: GATA3$^+$) were analyzed in eight mice per group. **l, m** Colon samples from two mice were pooled and Ly6C$^{+/-}$Ly6G$^-$ monocytes and Ly6C$^+$Ly6G$^+$ neutrophils (PMN) were analyzed by flow cytometry (6–8 samples per group). **i, l** Representative contour plots are shown. Numbers indicate percentages. **a–f, i–m** Data were pooled from two independent experiments with a total of 15–16 mice per group. Data show means + SEM; *$p \leq 0.05$; **$p \leq 0.005$; ***$p \leq 0.001$; ****$p \leq 0.0001$ (two-tailed Mann–Whitney $U$ test). Source data are provided as source data file.

(OTUs) with the Kruskal–Wallis test < 0.05 and LDA scores > 3.0 to identify statistically significant differences[41] between diseased NK$^{WT}$ and NK$^{\Delta c\text{-}FLIP}$ mice (Fig. 6f). In NK$^{WT}$ mice the genera *Prevotella* (OTU 8), *Muribaculum* (OTU 5; within the *Bacteriodales* S24-7), *Alistipes* (OTU 33; within the family *Rikenellaceae*) and two undefined OTUs (OTU 94 and OTU 76; within the family *Ruminococcaceae*) relatively expanded in the course of DSS colitis (Fig. 6f). In feces of NK$^{\Delta c\text{-}FLIP}$ mice the frequencies of bacteria belonging to the genera *Turicimonas* (OTU 75; within the *Alcaligenaceae*), *Turicibacter* (OTU 2; within the *Erysipelotrichaceae*), *Bacteroides* (OTU 12; within *Bacteroidaceae*) and a bacterium belonging to an ambiguous taxon (OTU 61; within the *Coriobacteriaceae*) were significantly increased. Elevated levels of *Turicibacter* and *Bacteroides* correlate positively with inflammation and tumor development in mice[42]. Moreover, expansion of *Alcaligenaceae* is frequently observed in Crohn's disease patients and drives systemic inflammation in mice[43]. Hence, the lack of NKp46$^+$ ILC in NK$^{\Delta c\text{-}FLIP}$ mice is associated with increased disease severity (Fig. 5a–f), altered cytokine expression (Fig. 5g, h) and the subsequent expansion of potentially pathogenic bacteria (Fig. 6a–f).

**IL-7R-dependent NKp46$^+$ ILC3 do not affect acute colitis.** In DSS-treated mice, NKp46$^+$ ILC3 are an important source of tissue-protective IL-22 and their deletion in T/B-deficient RAG$^{-/-}$ mice is associated with increased disease severity[8]. We therefore hypothesized that the lack of NKp46$^+$ ILC3 was responsible for disease aggravation in NK$^{\Delta c\text{-}FLIP}$ mice (Figs. 4a-h and 5a-f). In order to eliminate NKp46$^+$ ILC3 from T/B-competent mice, we exploited the strict IL-7R-dependence of NKp46$^+$RORγt$^+$ ILC3 (Fig. 3l). For this purpose, conditional IL-7R (IL-7R$^{fl/fl}$) knockout mice were crossed to NKp46$^{iCre}$-transgenic mice to generate NK$^{\Delta IL\text{-}7R}$ mice. As compared to NK$^{WT}$ mice, SI-derived helper-like NKp46$^+$CD90$^+$ ILC from NK$^{\Delta IL\text{-}7R}$ mice hardly expressed any IL-7Rα (CD127) while non-targeted NKp46$^-$CD90$^+$ ILC still expressed it at normal levels. As expected, NKp46$^+$CD90$^-$ cNK expressed least IL-7R in both mouse strains[32] (Fig. 7a). Next, we determined the composition of the ILC pool in the SI of NK$^{WT}$ and NK$^{\Delta IL\text{-}7R}$ mice. NK$^{\Delta IL\text{-}7R}$ mice were nearly completely devoid of NKp46$^+$RORγt$^+$ ILC3 (Fig. 7b, c), while numbers of NKp46$^+$Tbet$^+$ ILC1 (Fig. 7c), NKp46$^+$Eomes$^+$ cNK (Fig. 7b, c), and all NKp46$^-$ ILC subsets (Fig. 7c) appeared normal compared to NK$^{WT}$ mice.

Next, we tested whether the lack of NKp46$^+$RORγt$^+$ ILC3 affects the outcome of acute DSS colitis. For this purpose, NK$^{WT}$ and NK$^{\Delta IL\text{-}7R}$ mice were treated with DSS for 5 days and disease parameters were determined. As shown in Fig. 7d, relative body weight did not differ between both mouse strains. Altogether, our results demonstrate that the lack of NKp46$^+$RORγt$^+$ ILC3 does not affect the course of DSS-induced colitis.

**cNK cells ameliorate acute intestinal inflammation.** Since NKp46$^+$ ILC3 did not affect the course of DSS-induced colitis

(Fig. 7d), we hypothesized that NKp46$^+$ ILC1 deficiency accounted for disease aggravation in NK$^{\Delta c\text{-}FLIP}$ mice (Figs. 4a–h and 5a–f). This assumption was supported by the fact that the mild course of disease in NK$^{WT}$ mice correlated with the accumulation of c-FLIP-dependent Ly49C$^+$ cNK (Supplementary Fig. 3). In order to elucidate whether cNK are involved in disease modulation, B6 mice were treated with cNK-depleting anti-Asialo-GM1 (αAsialo) or control antibody. As shown in Supplementary Fig. 4A, αAsialo-treated mice suffered more from DSS-induced colitis than control animals. Since the efficacy of cNK depletion may vary between target tissues, we analyzed the frequencies and absolute numbers of colonic ILC. Anti-Asialo-treated mice contained significantly fewer NKp46$^+$NK1.1$^+$ ILC in the inflamed colon (Supplementary Fig. 4B, F, I). This reduction was mainly due to the depletion of NKp46$^+$NK1.1$^+$Eomes$^+$ cNK (Supplementary Fig. 4C, G, J) and correlated with elevated frequencies of neutrophils in peripheral blood (Supplementary Fig. 4E), similar to what we had observed in the inflamed colon of NK$^{\Delta c\text{-}FLIP}$ mice (Fig. 4l and m). On the contrary, frequencies of NKp46$^+$Eomes$^-$ ILC1 (Supplementary Fig. 4C) and NKp46$^{+/-}$RORγt$^+$ ILC3 (Supplementary Fig. 4D, H) remained largely unaffected by αAsialo treatment, whereas absolute numbers of the latter were significantly reduced (Supplementary Fig. 4K). Hence, antibody-mediated side effects on other ILC subsets could not be formally excluded.

As a result of NKp46-specific Eomes inactivation, cNK development is strongly impaired while other ILC remain largely unaffected[44–46]. We therefore generated cNK-deficient NK$^{\Delta Eomes}$ (NKp46$^{iCre}$ × Eomes$^{fl/fl}$) mice to further validate the anti-inflammatory function of cNK. NK$^{\Delta Eomes}$ mice and control animals were treated with DSS for 5 days. Similar to NK$^{\Delta c\text{-}FLIP}$ (Fig. 4a–d) and αAsialo-treated B6 mice (Supplementary Fig. 4A), disease severity was significantly increased in NK$^{\Delta Eomes}$ mice as compared to NK$^{WT}$ controls (Fig. 8a–d). In accordance with a more pronounced inflammatory response, colon weight of NK$^{\Delta Eomes}$ mice was increased (Fig. 8e) while its length was decreased (Fig. 8f, g). Importantly, we confirmed the selective loss of Eomes$^+$ cNK, but not helper-like ILC1, in the inflamed colon of NK$^{\Delta Eomes}$ mice (Fig. 8h, i). This correlated with the accumulation of neutrophils (Fig. 8j, k) similar to what we had observed for NK$^{\Delta c\text{-}FLIP}$ mice (Fig. 4l, m). Thus, cNK depletion, either by αAsialo treatment (Supplementary Fig. 4) or NKp46-specific Eomes inactivation (Fig. 8), is sufficient to aggravate disease. Hence, our results strongly suggest that cNK act as immune modulators in the inflamed colon with anti-inflammatory properties superior to all other NKp46$^+$ ILC.

**Discussion**

Immune cells are equipped with a multitude of cytokine receptors allowing them to sense changes in their environment and adapt activation as well as survival thresholds accordingly[13,47]. How cytokine signals are translated into ILC-specific developmental/survival programs is largely unknown. IL-15 is an environmental

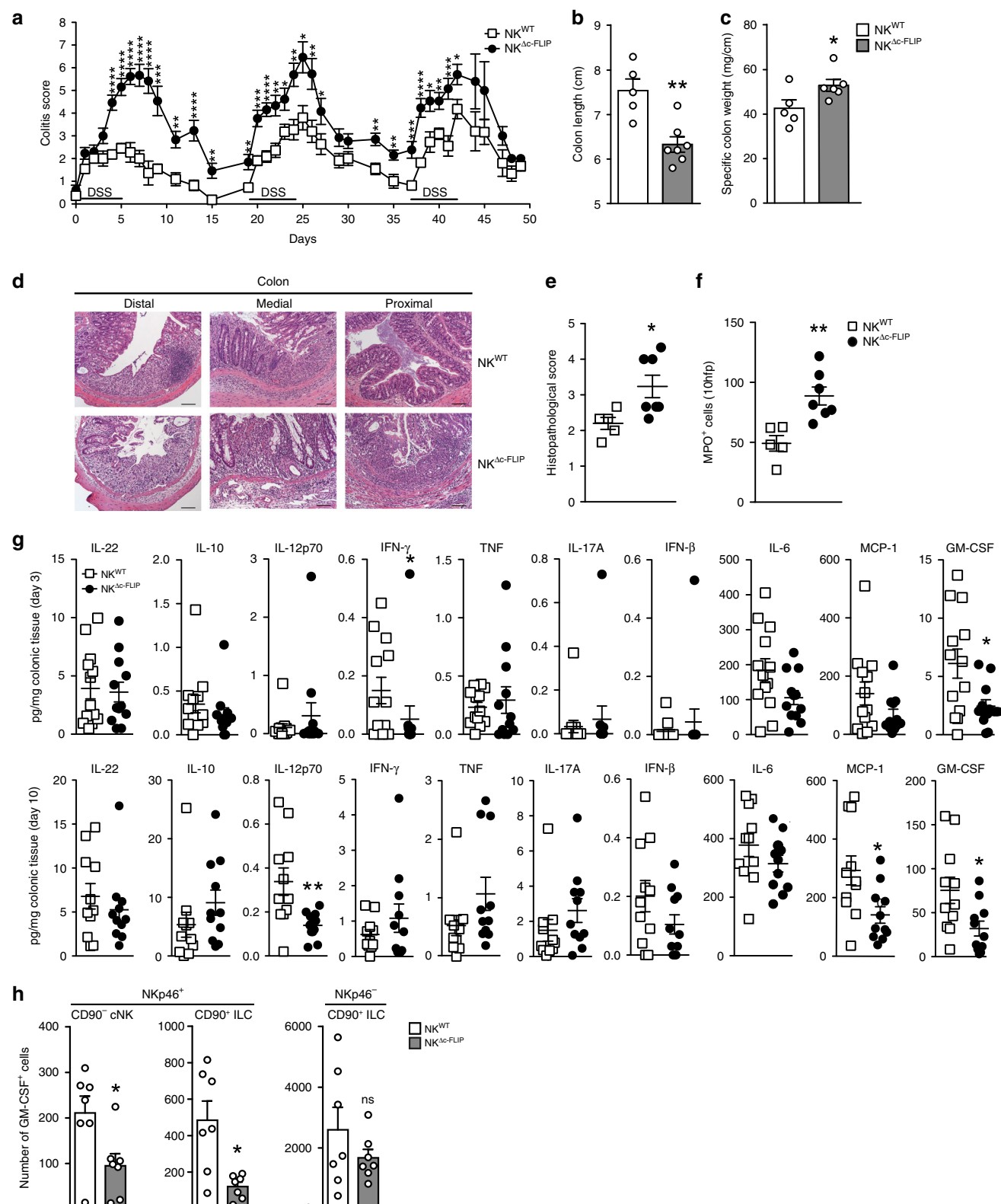

factor that is indispensable for ILC1/cNK development[25,26]. Previous studies demonstrated that IL-15 withdrawal impairs cNK metabolism[48] and activates the intrinsic apoptosis pathway[15]. In the absence of IL-15 cNK fail to downregulate the proapoptotic molecule Bim. At the same time, the induction of the anti-apoptotic Mcl-1 is impaired and IL-15-deprived cNK undergo apoptosis[15].

Here, we provide evidence for an additional mode of ILC-specific, non-redundant IL-15 action. We identify c-FLIP as a target of STAT5-dependent IL-15 signaling. c-FLIP is a master regulator of the extrinsic apoptosis pathway[20] and protects immune and non-immune cells from caspase 8-dependent apoptosis[21,22]. Usually, this type of apoptosis is triggered by DLs such as TNF or TRAIL, which are produced at high levels by

**Fig. 5 The lack of c-FLIP-dependent NKp46[+] ILC is associated with the early onset of chronic intestinal inflammation. a–h** NK[WT] and NK[Δc-FLIP] littermates were treated with DSS for 5 days followed by a 14-day recovery phase on normal drinking water. This treatment was repeated twice. **a** Shown are representative results (mean colitis score ± SEM) of one out of two independent experiments with 11–13 mice per group. **b–f** At day 42, **b** colon length and **c** specific colon weight were determined, **d–f** tissue samples from distal, medial, and proximal colon were collected and **d, e** stained with H&E (scale bars = 100 μm) to determine **e** histopathological scores (HPS). **f** For each colon segment, myeloperoxidase (MPO)[+] cells were enumerated in 10 high power fields (magnification ×400). **e, f** Each symbol represents mean values for individual mice after pooling results from the three colon segments. **b, c, e, f** Data show mean +/± SEM for the respective experimental groups. **b–f** 5–7 mice were analyzed. **g** At experimental days 3 (upper row) and 10 (lower row), colon samples were isolated and incubated for 24 h. Cytokine levels were determined in culture supernatants and normalized to tissue mass (n = 11–13/group). **h** At day 5, frequencies of GM-CSF-producing cells isolated from the colonic lamina propria were determined by flow cytometry. After gating on Lin[−]CD45[+] cells, GM-CSF production was analyzed for NKp46[+]CD90[−] cNK as well as NKp46[+/−]CD90[+] helper-like ILC. Pooled results (mean + SEM) from four independent experiments with a total of seven colon samples (pooled from 2 to 3 mice) per group are shown. Statistical significances were determined using **a, e, f, g, h** two-tailed Mann–Whitney U and **b, c** two-tailed unpaired Student's t test (*p ≤ 0.05; **p ≤ 0.005; ***p ≤ 0.001; ****p ≤ 0.0001; ns = not significant). **a–c** and **e–h** Source data are provided as source data file.

mature ILC1[29]. To prevent their DL-induced premature death in the course of an immune response c-FLIP is required[30]. Our results demonstrate that immature ILC already require c-FLIP in the very early phase of their development. This appears to be due to the fact that developing ILC up-regulate effector genes such as *Ifng* and *Tnf* already at the SP stage. This is paralleled by DR up-regulation, particularly of TNF-R1, thus rendering BM ILC sensitive to TNF-induced apoptosis. Consequently, only c-FLIP-competent ILC precursors developed into DP ILC in vitro. Importantly, the blockade of TNF partially restored the generation of c-FLIP-deficient DP ILC. Whether TNF production by developing ILC is a cell autonomous process or whether it is influenced by environmental factors remains to be shown.

T/B-deficient mice lacking NKp46[+] ILC3 succumb to *C. rodentium* infection. On the contrary, their T/B-competent counterparts control infection[9,10]. This demonstrates that adaptive immune cells and NKp46[+] ILC have redundant functions[9,10], a conclusion that is supported by human data[11]. However, NKp46[+] ILC3 are indispensable for the maintenance of cecal homeostasis[9]. Hence, the relative contribution of NKp46[+] ILC to the modulation of immune responses appears to vary in a context-dependent fashion. This is further exemplified by apparently opposing results obtained in mouse models of intestinal inflammation. While NKp46[+] ILC3 protect T/B-deficient mice from DSS-induced colitis[8], they promote αCD40-induced colitis in T-deficient mice[10]. Whether and how NKp46[+] ILC affect the severity of DSS-induced colitis in T/B-competent mice had not been studied in detail before. To address this issue, we used ILC1/3-deficient, T/B-competent NK[Δc-FLIP] mice. Our data clearly demonstrate that NKp46[+] ILC are indispensable for disease control. Neither T or B lymphocytes nor NKp46[−] ILC were able to compensate for the lack of NKp46[+] ILC in NK[Δc-FLIP] mice. Importantly, with the help of NK[ΔIL-7R] mice, we could exclude NKp46[+] ILC3 as major regulators of acute DSS-induced colitis emphasizing the importance of NKp46[+] ILC1. In DSS-treated NK[Δc-FLIP] mice, ILC-deficiency correlated with increased disease scores, altered cytokine profiles and the accumulation of neutrophils in the inflamed colon. Our data suggest that this neutrophil-associated effect was due to the lack of cNK, which can limit the recruitment and pro-inflammatory function of neutrophils[49]. This anti-inflammatory function of cNK is NKG2A-dependent as shown by the fact that anti-NKG2A antibodies block cNK–neutrophil interactions in vitro and exacerbate DSS-induced colitis in vivo. Based on these observations, a dominant regulatory role of cNK in DSS-induced colitis was postulated[49]. However, NKG2A is not only expressed by cNK but also by, e.g. ILC1[26] and CD8[+] T cells[50]. Hence disease exacerbation in response to anti-NKG2A treatment[49] may involve multiple NKG2A[+] immune cell types. However, as we have shown here, (i) the mild course of disease in NK[WT] mice correlated with the accumulation c-FLIP-dependent Ly49C[+] cNK, (ii) αAsialo-GM-mediated cNK depletion aggravated disease in B6 mice and, above of all, cNK-deficient NK[ΔEomes] mice were by far more sensitive to DSS-induced colitis as compared to controls. Of note, disease aggravation was comparable for cNK-deficient NK[ΔEomes] and NK[Δc-FLIP] mice lacking all NKp46[+] ILC. Hence, our findings support the view that cNK have a dominant anti-inflammatory function in DSS-induced colitis, which results, at least partially, from the restriction of neutrophil infiltration and function in the inflamed colon[51]. However, it is important to emphasize that the cell surface molecules used to define neutrophils in our experimental system are used to define myeloid-derived suppressor cells (MDSCs) in others[52]. In addition to their nearly identical cell surface phenotype, MDSCs and neutrophils share functional features as well. For example, and contrary to their well-known anti-inflammatory functions in tumor models[53], MDSCs were shown to promote DSS-induced colitis[54] similar to neutrophils[49]. Hence, our experimental approach does not allow us to define the relative contribution(s) of MDSCs and neutrophils to disease progression. Based on the complex intercellular interactions driving colitis[55] we also cannot exclude that cNK deficiency promotes pro-inflammatory functions of other cell types.

NK[Δc-FLIP] mice developed maximum disease scores after only a single phase of DSS administration. Furthermore, they failed to recover completely after the first DSS cycle and showed maximum disease scores after each additional DSS phase. This early establishment of chronic disease in NK[Δc-FLIP] mice correlated with the prolonged reduction of tissue-protective GM-CSF and lower numbers of GM-CSF[+]NKp46[+] ILC including cNK. Hence, the protective effect of cNK may not only rely on the suppression of neutrophil effector functions but also on the production of tissue-protective GM-CSF. It is important to stress that no other immune cell type, including NKp46[+] ILC1/3, is able to compensate for the early protective and regenerative effects of NKp46[+] cNK. This may, at least partially, rely on the disease-related control of the commensal microbiota. Of note, the lack of NKp46[+] ILC in NK[Δc-FLIP] mice did not have a significant impact on the composition of the commensal microbiota prior to DSS-induced colitis. This argues against a direct impact of NKp46[+] ILC on the establishment of the microbiota under homeostatic conditions in T/B-competent mice. Nevertheless, after establishment of chronic colitis the composition of the commensal microbiota changed in a NKp46[+] ILC-dependent fashion. For example, *Coriobacteriaceae*, *Alcalignaceae*, *Erysipelotrichaceae*, and *Bacteroidaceae* were more abundant in NK[Δc-FLIP] mice, while *Rikenellaceae*, *Bacteroidales* S24-7, *Prevotellaceae*, and *Ruminococcaceae* were enriched in NK[WT] mice. Commensal dysbiosis is frequently observed in different disease models and the altered abundance of certain bacteria has disease-modulating effects[37,38]. For example, the

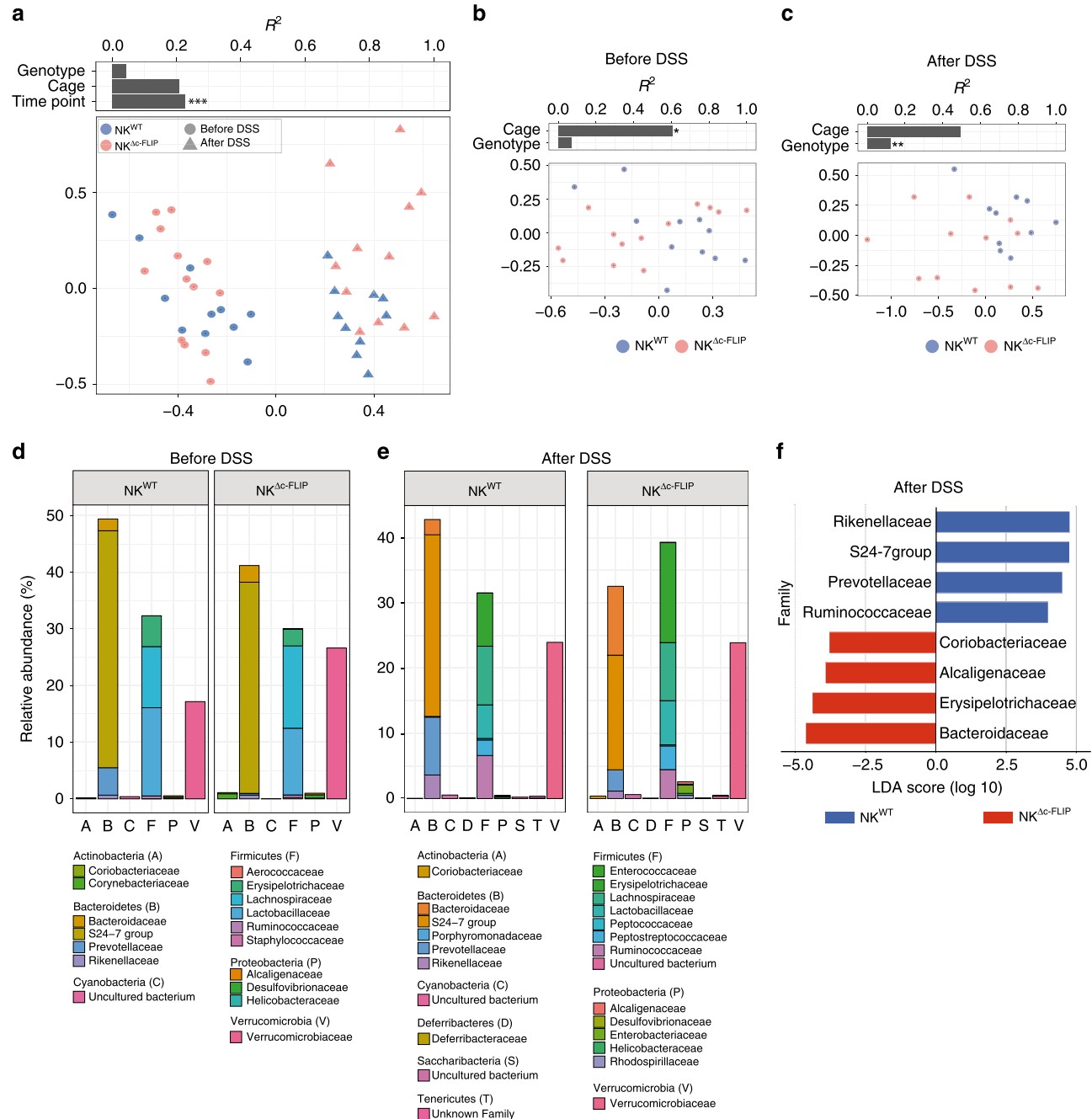

**Fig. 6 The lack of c-FLIP-dependent NKp46⁺ ILC is associated with disease-related alterations in microbiome composition. a–f** Feces samples of NK$^{WT}$ and NK$^{\Delta c\text{-FLIP}}$ littermates (Fig. 5) were collected before DSS administration (before DSS) and at the end of the observation period (after DSS, d49). Whenever possible, mice shared cages from birth on and throughout the experiments. Analyzed samples ($n = 10$–13/group and time point) had a minimum sequencing depth of 1000 reads and a mean sequencing depth of 19,298.3 ± 7781.349 reads (mean ± SD). **a–c** Non-metric multidimensional scaling (NMDS) ordination analysis of fecal microbiota composition was performed using Bray–Curtis distances grouped by **a** genotype and treatment or **b**, **c** genotype **b** before and **c** after DSS-induced colitis. Individual effect size of tested covariates is indicated. **a–c** To calculate the variance explained by individual factors such as genotype, cage effect, and treatment a permutational multivariate analysis of variance (ADONIS) was used. *$p < 0.05$; **$p < 0.01$; ***$p < 0.001$. A significant effect was dedicated when $p < 0.05$ and $R^2 > 0.10$ (equivalent to 10% of explained variance). **d**, **e** Relative abundance of the average microbiome composition **d** before and **e** after DSS administration was determined on the family level. Phylum and families are indicated. **f** Statistically significant differences on family levels in fecal microbiota composition between NK$^{WT}$ and NK$^{\Delta c\text{-FLIP}}$ mice. Data were analyzed using linear discriminant analysis (LDA) effect size (LEfSe) method (Kruskal–Wallis test with $p < 0.05$ and LDA scores > 3.0). Data are displayed as bar plot ranked according to their effect size and associating them to the genotype of the mice. **a–f** Source data are provided as source data file.

overabundance of *Bacteroides* has been associated with increased rates of inflammation and tumor development[42]. Moreover, the expansion of bacteria from the family *Alcaligenaceae* drives inflammation in ILC-depleted RAG1$^{-/-}$ mice[43]. On the contrary, the enhanced relative abundance of *Prevotella* in diseased NK$^{WT}$ mice may have beneficial effects, since it correlates inversely with inflammation in a mouse model of inflammation-induced carcinogenesis[42]. Altogether, these results demonstrate that the

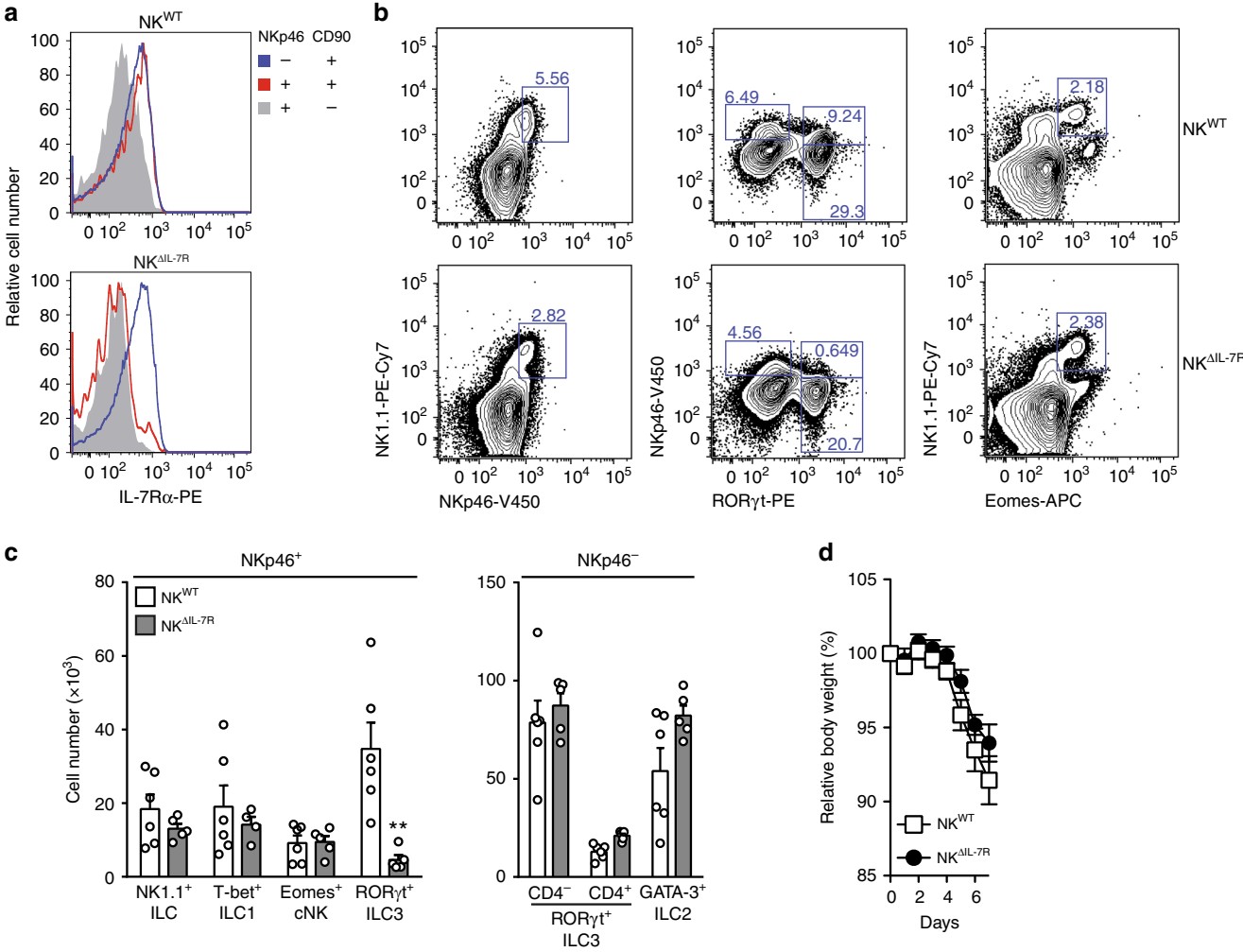

**Fig. 7 IL-7R-dependent ILC3 do not affect acute colitis. a–c** Small intestinal lamina propria leukocytes were isolated from NK$^{WT}$ and NK$^{ΔIL-7R}$ mice. Cells were analyzed by flow cytometry as described in Fig. 3. **a** Overlays show relative fluorescence intensities of IL-7Rα (CD127) for the indicated ILC. Data are representative for four mice per genotype. **b** Shown are representative contour plots and **c** pooled results (means + SEM) from two independent experiments with 5–6 mice per group. Four to six samples were analyzed. **d** To induce acute colitis, NK$^{WT}$ and NK$^{ΔIL-7R}$ mice were treated with DSS for 5 days as described in Fig. 4. Relative body weights were determined on a daily basis. Pooled results (means ± SEM) from two independent experiments with 17–18 mice per group are shown. **c, d** Source data are provided as source data file; *$p ≤ 0.05$; **$p ≤ 0.005$; ***$p ≤ 0.001$; ****$p ≤ 0.0001$ (two-tailed Mann–Whitney $U$ test).

expansion of disease-promoting microbes, such as *Proteobacteria*, i.e. *Alcaligenaceae*, occurs selectively in DSS-treated NK$^{Δc-FLIP}$ mice. This indicates that c-FLIP-dependent NKp46$^+$ ILC modulate the composition of the commensal microbiota under inflammatory conditions but not prior to disease. Whether alterations of the commensal microflora are a cause or a result of inflammation remains an important question for the future.

In summary, we identify the anti-apoptotic molecule c-FLIP as a target of cytokine-induced STAT5 activation, which is indispensable for the development of IL-15- and IL-7-dependent NKp46$^+$ ILC1 and ILC3, respectively. Furthermore, we provide evidence that cNK, but not ILC1/3, restrict the degree of intestinal inflammation correlating with specific alterations in the commensal microbiota. Hence, c-FLIP-dependent NKp46$^+$ cNK protect T/B-sufficient mice from intestinal inflammation, a function that cannot be replaced by any other immune cell.

## Methods
**Mice.** NKp46$^{iCre}$ [19], c-FLIP$^{fl/fl}$ [23], STAT5$^{fl/fl}$ [56] (provided by L. Hennighausen), IL-15$^{−/−}$ [25], IFN-γ reporter mice (Great)[57], RAG1$^{−/−}$[58], IL-7R$^{−/−}$[59], and RAG1$^{−/−}$ × IL-7R$^{−/−}$ were maintained under specific pathogen-free conditions at

the central animal facility of the Medical Faculty of the Otto-von-Guericke-University Magdeburg. IL-7R$^{fl/fl}$ (stock no. 022143)[60], Eomes$^{fl/fl}$ (stock no. 017293)[61], and IL-15Rα$^{−/−}$ mice (stock no. 003723)[31] including appropriate control mice were purchased from The Jackson Laboratory. Whenever possible, control littermates were used and cohoused (age 8–16 weeks). Both sexes were considered equally. Experimental procedures were approved by the relevant animal experimentation committee and performed in compliance with international and local animal welfare legislations (Landesverwaltungsamt Sachsen-Anhalt, permit numbers AZ 42502-2-1202 and AZ 42502-2-1521 Uni MD).

**Cell isolation.** Mice were used euthanized by isoflurane inhalation overdose. Single-cell suspensions of spleen were prepared by forcing the organs through metal sieves. BM leukocytes were flushed from tibia, femur, and coxal bone using PBS/2 mM EDTA (ROTH). To lyse erythrocytes, cell suspensions were incubated with ACK lysis buffer for 90 s and subsequent addition of RPMI 1640 (Biochrome) with 10% FCS (PAN Biotech). Cells resuspended in PBS/2 mM EDTA were filtered through 40 µm cell strainers (Corning, Durham, NC, USA).

LPL were isolated from different parts of the intestine according to published protocols[2,62]. Briefly, residual mesenteric fat and Peyer's Patches were carefully removed. Intestinal tissue was opened longitudinally and washed two to three times in ice-cold PBS with 100 U/ml penicillin/streptomycin (P/S; GIBCO Life Technologies). The epithelial layer and containing intraepithelial leukocytes were removed by two incubation steps in Hank's balanced salt solution without Ca$^{2+}$ and Mg$^{2+}$ (HBSS; Biochrom) supplemented with 5 mM EDTA, 10 mM HEPES and P/S for 20 min at RT with slow rotation (100 rpm) and subsequent vortexing

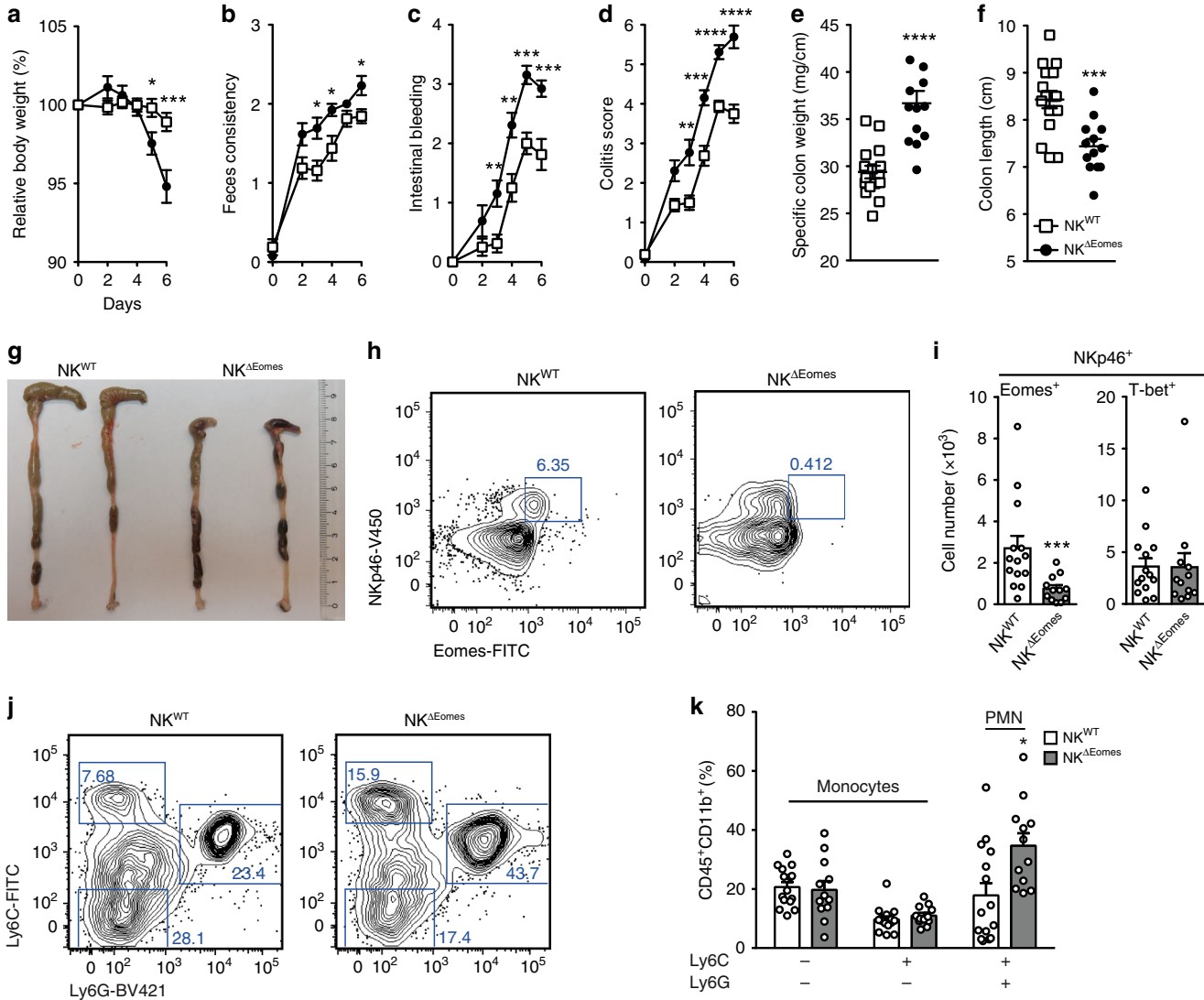

**Fig. 8 Conventional NK cells ameliorate acute colitis. a–k** $NK^{WT}$ and $NK^{\Delta Eomes}$ mice were treated with DSS for 5 days as described in Fig. 4. **a** Relative body weight, **b** feces consistency, and **c** the degree of intestinal bleeding were measured on a daily basis to calculate **d** the overall colitis score. **e–g** At day 7, **e** the specific weight and **f**, **g** the length of the colon were determined. **g** Shown are representative photographs of colon and cecum (scale unit = cm). **h**, **i** Colonic lamina propria leukocytes were analyzed by flow cytometry at day 7. Frequencies of $Lin^-CD45^+$ $NKp46^+$ ILC1 (T-bet$^+$) and cNK (Eomes$^+$) and **j**, **k** $CD45^+CD11b^+$ $Ly6C^{+/-}Ly6G^-$ monocytes as well as $Ly6C^+Ly6G^+$ neutrophils (PMN) were determined by flow cytometry. **i** Absolute cell numbers were calculated. **h**, **j** Representative contour plots are shown. Numbers indicate percentages. **a–f**, **i** and **k** Data were pooled from two independent experiments with a total of 12–16 mice per group. Data show means +/± SEM. Source data are provided as source data file; $^*p \leq 0.05$; $^{**}p \leq 0.005$; $^{***}p \leq 0.001$; $^{****}p \leq 0.0001$ (two-tailed Mann–Whitney $U$ test). **i**, **k** 12–15 tissue samples were analyzed.

for 15 s. Tissue pieces were washed in PBS P/S, minced and incubated 2–3 times for 20 min at 37 °C in digestion solution containing 4% FCS, 0.25 mg/ml each of collagenase D (Roche), DNaseI (Sigma-Aldrich), and Dispase II (Roche) followed by vigorous vortexing and passage through 40 µm cell strainers. Leukocytes were enriched by density gradient centrifugation at $600 \times g$ for 20 min at RT.

**Flow cytometry**. The following reagents were purchased from BioLegend or eBiosciences: anti-mouse CD3 (145-2C11), CD4 (RM4-5), CD5 (53-7.3), CD8a (53-6.7), CD11b (M1/70), CD11c (N418), CD19 (1D3), CD45 (30-F11), CD120a (TNF-RI; 55R-286), CD122 (TM-β1), Eomes (Dan11mag), F4/80 (BM8), GM-CSF (MP1-22E9), IFN-γ (XMG1.2), Ly6C/G/Gr-1 (RB6-8C5), Ly6G/Gr1 (1A8), Ly6C (HK1.4), MCP-1 (2H5), NK1.1 (PK136), T-bet (4B10), Ter119 (TER-119), TNF (MP6-XT22), TRAIL (N2B2), armenian hamster IgG1 (RTK2073), donkey-anti-rabbit IgG (Poly4064), rat IgG1, κ (RTK2071), rat IgG2a, κ (RTK2758), and Zombie viability dye. Anti-mouse CD16/32 (2.4G2; ATCC® HB-197™) was produced in our laboratory. Anti-mouse CD11b (M1/70), GATA-3 (TWAJ), NKp46 (29A14), pSTAT5 pY694 (47/Stat5), RORγt (B2D), and Streptavidin-FITC were purchased from BD Biosciences. Polyclonal rabbit-anti-c-FLIP was purchased from Abcam. Anti-mouse Ly49A (A1), Ly49C/I (REA253), and Eomes (REA116) were

purchased from Miltenyi Biotec. Further antibody information is provided in Supplementary Table 1.

Single cell suspensions of spleen, BM as well as LPL were stained with fluorochrome-labeled antibodies for 30 min at 4 °C after blocking Fc receptors with anti-CD16/32. For detection of cytoplasmic antigens, cells were incubated in fixation buffer (BioLegend) for 30 min at 4 °C, permeabilized and stained with the appropriate antibodies in permeabilization wash buffer (BioLegend) for 30 min at 4 °C. For staining of nuclear antigens, cells were permeabilized and fixed using the Foxp3 staining buffer set (eBioscience) according to the manufacturer's instructions. Peripheral blood leukocytes were stained in whole blood samples for 30 min at RT, followed by erythrocyte lysis and fixation using BD FACS™ lysing solution (BD Biosciences). Samples were measured on a LSR Fortessa and FACSCanto flow cytometer (Becton Dickinson) and analyzed by FlowJo software (FlowJo LLC).

**In vitro stimulation of splenic ILC**. Splenocytes were isolated from C57BL/6 mice and cultured for 16 h in RPMI (with 10% FCS and 100 U/ml P/S) supplemented with IL-15 (10 ng/ml, Peprotech), with rmIL-15 (10 ng/ml) and pimozide (5 µM, Calbiochem) or were left untreated. Lineage negative ($CD3^-$, $CD4^-$, $CD8^-$, $CD19^-$,

Gr1⁻, Ter119⁻) CD122⁺NK1.1⁺NKp46⁺ ILC were analyzed by flow cytometry for their expression of c-FLIP and phosphorylation of STAT5.

**Detection of apoptotic cells.** Apoptotic cells were visualized by flow cytometry by using the FAM-FLICA Caspase 3/7 assay kit from ImmunoChemistry Technologies according to the manufacturer's protocol. The cell permeable, fluorescent inhibitor probe FAM-DEVD-FMK irreversibly binds to active caspases 3 and 7.

**In vitro culture of ILC precursors.** BM-derived ILC precursors were enriched by depletion of Ter119⁻, CD19⁻, and LyG/Gr1-positive cells using the respective biotinylated antibodies (Ter119, CD19, and Ly6G/Gr1; clones as indicated above) and anti-biotin Dynabeads (Invitrogen) according to the manufacturer's recommendations. Next, the Ter119/CD19/Gr1-depleted cell suspension was labeled with antibodies against CD3, CD4, CD8, CD122, and NK1.1 and FITC-conjugated streptavidin. Lineage negative (Ter119⁻, CD19⁻, Ly6G/Gr1⁻, CD3⁻, CD4⁻, CD8⁻) CD122⁺NK1.1⁻ BM ILC precursors were enriched (on average 42%) by fluorescence-activated cell sorting (FACS Vantage DIVA, FACS Aria III, Becton Dickinson). ILC precursors were co-cultured with irradiated (4 Gy) OP9 cells (ATCC® CRL-2749™) in α-MEM medium (GIBCO Life Technologies) containing 10% FCS, 20 ng/ml SCF (Peprotech), 20 ng/ml Flt-3 ligand (Peprotech), 20 ng/ml IL-7 (eBioscience), 20 ng/ml IL-15 (Peprotech), 150 μM 1-thioglycerol (Sigma-Aldrich), 2 mM L-glutamine (GIBCO Life technologies), 1 mM Na-pyruvate (GIBCO Life technologies), 100 U/ml penicillin/streptomycin, and 1 x primocin (Amaxa) for 9 days. After 4 days of co-culture, 50% of medium was replaced. Two days later cells were moved to a fresh OP9 feeder cell layer. At day 9 of co-culture cells were analyzed by flow cytometry. To block TNF-, FASL- or TRAIL-mediated signaling, 10 μg/ml of anti-TNF (MP6-XT22), anti-FASL (3C82; Enzo), or anti-TRAIL (N2B2), respectively, were added to the co-cultures every 2.5 days. Isotype-matched control antibodies were used as controls. Blocking/control reagents were purchased from BioLegend.

**Quantitative real-time PCR (RT-qPCR).** FACS-sorted splenic ILC (CD3⁻NKp46⁺ NK1.1⁺) were stimulated with IL-15 (50 ng/ml) for 16 h or were left untreated. ILC progenitors were sorted using MoFlo Sorter (Beckman Coulter) from BM extracts. RNA isolation from cell cultures and primary cells was performed using QIAshredder and RNeasy Mini Kit (QIAGEN) according to manufacturer's protocol. cDNA was synthetized using the cDNA Synthesis Kit (Life technologies) according to manufacturer's instructions. The cDNA served as a template for the amplification by real-time PCR, using SYBR Green (Roche) in a LightCycler 96 instrument (Roche). Ubiquitin-conjugating enzyme E2D 2A (UCE) was used as reference gene. Primers: UCE fwd: 5′-AAGAGAATCCACAAGGAATTGAATG-3′; UCE rev: 5′-CAACA GGACCTGCTGA ACACTG-3′. CD95 (FAS) fwd: 5′-GCAGACAUGCUGUGGA UCUG-3; CD95 (FAS) rev: 5′-UCGGAGAUGCUAUUAGUACCUUGAG-3. DR5 (TRAIL-R2) fwd: 5′-CCCUGAGAUCUGCCAGUCAU-3′; DR5 (TRAIL-R2) rev: 5′- UGGGGGUACAGGAAGUCAGU-3′. TNF-R1 fwd: 5′-GAAAGUAUGUCCAUUC UAAGAACAA-3′; TNF-R1 rev: 5′-AGUCACUCACCAAGUAGGUUCCUU-3′. c-FLIP$_L$ fwd: 5′-GCAGAAGCUCUCCCAGCA-3; c-FLIP$_L$ rev: 5′-UUUGUCCAUGAG UUCAACGUG-3′. c-FLIP$_R$ fwd: 5′-UCCAGAAGUACACCCAGUCCA-3′; c-FLIP$_R$ rev: 5′-CACUGGCUCCAGACUCACC-3′.

**DSS-induced colitis.** Acute colitis was induced by administration of 1.8–2.5% (w/v) dextran sulfate sodium (DSS; MW 36,000–50,000; ICN Biomedicals) ad libitum via the drinking water for 5 consecutive days. Chronic colitis was induced by repeated DSS administration (1.6%) for 5 days interrupted by 14-day periods on normal drinking water. Total body weight and feces samples were monitored. Feces consistency was evaluated to determine the degree of diarrhea. Furthermore, feces samples were analyzed for occult blood to determine the degree of intestinal bleeding. Occult blood was detected using hemoCARE test (Care diagnostic). Disease activity score was calculated by summarizing the scores for weight loss, feces consistency, and bleeding[63].

**Histological analysis.** Tissue samples from distal, medial, and proximal colon (0.5–1 cm) were fixed in 4% formalin and embedded in paraffin. 1–2 μm-thick sections were cut, dewaxed, and histochemically stained with hematoxylin & eosin (Merck). Histomorphological changes were evaluated in (HE)-stained tissue sections and images (original magnification ×100) were taken by standard light microscopy using an AxioImager Z1 microscope (Carl Zeiss MicroImaging, Jena, Germany). Histopathological evaluation was performed in a blinded manner using a standardized histologic score ranging from 0 to 6[64].

**Analysis of colonic cytokine production.** Samples of ~1 cm in length were isolated from the proximal, medial, and distal part of each colon. Samples were cleaned, opened longitudinally, and weighed. These colonic tissue explants were cultured in 500 μl complete RPMI 1640 supplemented with 2% FCS and 100 U/ml P/S in 24-well flat-bottom plates at 37 °C/5% CO₂ for 24 h. Culture supernatants were harvested and subsequently centrifuged at 10,000 × g for 10 min to remove debris. Supernatants were stored at −80 °C. Cytokine levels were measured using the Legendplex Mouse inflammation panel (Biolegend) or IL-22-ELISA

(Affimetrix/eBioscience) according to the manufacturer's instructions. Cytokine levels were normalized to tissue mass and expressed as picogram of cytokine per microgram of tissue. Normalized cytokine data from the three colonic tissue explants per mouse were pooled.

For the flow cytometric analysis of cytokine production, colon samples from 2 to 3 mice were pooled, LPL were enriched and cultured subsequently for 5 h in the presence of 20 ng/ml phorbol myristate acetate (PMA)/1 μg/ml ionomycin. Brefeldin A (BD GolgiPlug™; BD Biosciences) was added for the last 4 h of culture. Stimulated LPL were washed and incubated with fluorochrome-labeled antibodies for 30 min at 4 °C in the dark. After washing, LPL were fixed using the Fixation buffer for intracellular staining procedures (Biolegend) for 15 min in the dark at RT. Samples were washed with PBS/2 mM EDTA and stored in PBS/0.5% (w/v) BSA/2 mM EDTA containing anti-CD16/32 overnight. Subsequently, cells were washed in 200 μl permeabilization buffer (Biolegend) and incubated with fluorochome-labeled anti-cytokine mAbs or isotype-matched control antibodies for 1 h at 4 °C. After washing cells were resuspended in PBS/2 mM EDTA, measured on a LSR Fortessa (Becton Dickinson) flow cytometer and analyzed by FlowJo software (FlowJo LLC).

**NK depletion.** C57BL/6 mice were injected i.p. with 50 μg of NK-depleting anti-Asialo GM1 (Affimetrix). Treatment was repeated every third day. A total of four injections were performed. Control animals were treated accordingly with 50 μg polyclonal rabbit control IgG (Sigma-Aldrich).

**Statistical analysis.** Experimental results were plotted and analyzed for statistical significance with Prism software 5 or 6 (GraphPad Software Inc.). Statistical significances were determined using two-tailed Mann–Whitney $U$ test, paired and unpaired Student's $t$ test or two-tailed Wilcoxon matched-pairs signed rank test (*$p \leq 0.05$; **$p \leq 0.005$; ***$p \leq 0.001$; ****$p \leq 0.0001$). Exact $p$ values are provided in Source data.

**DNA isolation.** Feces samples were collected and stored at −20 °C until processing for DNA-based 16S rRNA gene sequencing. DNA was extracted using an established phenol–chloroform-based method[65]. In short, 500 μl of extraction buffer (200 mM Tris (Roth), 20 mM EDTA (Roth), 200 mM NaCl (Roth), pH 8.0), 200 μl of 20% SDS (AppliChem), 500 μl of phenol:chloroform:isoamyl alcohol (PCI) (24:24:1) (Roth) and 100 μl of zirconia/silica beads (0.1 mm diameter) (Roth) were added to each feces sample. Samples were lysed and homogenized twice using a Mini-BeadBeater-96 (BioSpec) for 2 min. After centrifugation and additional extraction with PCI (24:24:1), DNA was precipitated using 500 μl isopropanol (J.T. Baker) and 0.1 volume of 3 M sodium acetate (AppliChem). Samples were incubated at −20 °C overnight and centrifuged at 4 °C at maximum speed for 20 min. The resulting DNA pellet was dried, resuspended in TE buffer (AppliChem) with 100 μg/ml RNase I (Sigma-Aldrich) and column purified (BioBasic Inc.) to remove PCR inhibitors.

**16S rRNA gene amplification and sequencing.** 16S rRNA gene amplification of the V4 region (F515/R806) was performed according to an established protocol previously described[40]. Briefly, DNA was normalized to 25 ng/μl and used for sequencing PCR with unique 12-base Golary barcodes incorporated via specific primers (obtained from Sigma-Aldrich). Using Q5 polymerase (New England Biolabs), PCR was performed in triplicates for each sample, using PCR conditions of initial denaturation for 30 s at 98 °C, followed by 25 cycles (10 s at 98 °C, 20s at 55 °C, and 20 s at 72 °C). After pooling and normalization to 10 nM, PCR amplicons were sequenced on an Illumina MiSeq platform via 250 bp paired-end sequencing (PE250).

**16S rRNA gene analysis.** Using Usearch8.1 software package (http://www.drive5. com/usearch/) the resulting reads were assembled, filtered, and clustered. In brief, merging was performed using -fastq_mergepairs –with fastq_maxdiffs 30 and quality controlling was performed with fastq_filter (-fastq_maxee 1), using a minimum read length of 200 bp and a minimum number of sequences per sample = 1000 as thresholds. Reads were clustered into 97% ID OTUs and representative sequences were determined by use of UPARSE algorithm[66]. Taxonomy classification was performed with the Silva database v128[67] and the RDP Classifier executed at 80% bootstrap confidence[68]. Resulting OTU absolute abundance table and mapping file were used for statistical analyses and data visualization using R statistical programming environment (R Core Team (2016)) package phyloseq[69].

Statistics were performed with 'R' version 3.3.0 (2016-05-03), (http://www. rproject.org) and the packages 'phyloseq'[69], and 'ggplot2'[70]. Non-parametric Mann–Whitney $U$ tests were considered as significant when $P$ values were lower than 0.05 after multiple testing correction (Benjamini–Hochberg false discovery rate correction). Permutational multivariate analysis of variance test (ADONIS) was used to assess the influence of different variables on the sample variability. The ADONIS tests were computed using 999 permutations and a resulting $R^2 > 0.1$ (effect size 10%) and $P$-values < 0.05 were considered as significant. LDA effect size (LEfSe) method was used to identify bacterial OTUs that explained differences between microbiota compositions[41]. Only OTUs with Kruskal–Wallis test < 0.05 and LDA scores > 3.0 were considered for analysis.

**Reporting summary**. Further information on research design is available in the Nature Research Reporting Summary linked to this article.

## Data availability

16S rRNA gene sequencing data have been deposited in the NCBI (Bioproject Database) under the accession number: PRJNA437582. The source data underlying Figs. 1a–d, 2a–f, 3e–k, 4a–f and i–m, 5a–c, and e–h, 6a–f, 7c and d, 8a–f, i and k and Supplementary Figs. 1, 2, 3a, and e, 4a and e–k are provided as a Source Data file. All other data that support the findings of this study are available from the corresponding author on reasonable request.

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

## Acknowledgements

We thank E. Denks, J. Giese, C. Kozowsky, J. Nichelmann, S. Schumann, and M. Berger for excellent technical assistance. We are grateful to Dr. Y.-W. He for providing c-FLIP[fl/fl] mice. This work was supported by the Deutsche Forschungsgemeinschaft Sonderforschungsbereich SFB854 (project A22) and DFG priority program 1937 (project SCHU 2326/2-1).

## Author contributions

A.A.K., I.R.D., T.S., I.S., and T.S. designed and supervised the study with the help of U.B. and K.D.; U.B., K.D., C.P.-S., L.O., A.W., L.K., R.L., R.J., F.R., and A.B. performed the experiments and analyzed the data; A.C.Z., C.R., and E.V. provided essential reagents; T.S. wrote the manuscript with the help of the other co-authors.

## Competing interests

The authors declare competing financial interests. The E.V. laboratory is supported by the European Research Council (ERC) under the "European Union's Horizon 2020 research and innovation program (Grant agreement no. 694502)", Agence Nationale de la Recherche, Innate Pharma, MSDAvenir, Ligue Nationale contre le Cancer (Equipe labelisée "La Ligue") and Marseille-Immunopole. E.V. is cofounder, shareholder, and employee of Innate Pharma.
