## [Peer Review File · Nature Communications]

Reviewers' comments:

Reviewer #1 (ILC, microbiota)(Remarks to the Author):

Bank et al. examined the role of c-FLIP molecule in NKp46+ ILCs and the role of NKp46+ ILCs in colitis. c-FLIP is an anti-apoptotic molecule protecting cells from death ligand-induced apoptosis. The expression of c-FLIP is induced by STAT5 downstream of IL-7 and IL-15, which are critical regulators of ILC differentiation. The authors deleted c-FLIP from NKp46+ cells by crossing c-FLIP<flox/flox> mouse with NKp46<iCre> mouse. c-FLIP<flox/flox>NKp46<iCre> mice have reduced number of NKp46+ ILC1 including cNK cells and NKp46+ ILC3. The authors also generated STAT5<flox/flox>NKp46<iCre> mice and showed similar results. c-FLIP<flox/flox>NKp46<iCre> mice exhibited higher susceptibility to DSS-colitis compared to WT mice. To examine the role of NKp46+ ILC3, the authors exploited the higher dependence of NKp46+ ILC3 on IL-7 signals compared to ILC1 and used IL-7R<flox/flox>NKp46<iCre> mice, which lack NKp46+ ILC3 but maintain most ILC1/cNK. These mice showed susceptibility to DSS similar to WT mice, suggesting that NKp46+ ILC1/cNK but not NKp46+ ILC3 play a protective role for DSS colitis. Depletion of cNK by anti-asialoGM1 antibody accelerated inflammation, indicating that cNK cells play a protective role. To examine the role of c-FLIP for differentiation of NKp46+ cells, the authors showed that the expression of death ligands such as TNF α and TRAIL increased during NK/ILC1 differentiation as c-FLIP. In vitro culture of NK1.1-NKp46- progenitor revealed that the differentiation of progenitor cells to NK1.1+ NKp46+ mature ILC1/NK cells was greatly impaired when progenitor from c-FLIP<flox/flox>NKp46<iCre> mice were used. Addition of anti-TNF α neutralizing antibody partially rescued ILC1/NK cell maturation in this in vitro culture system. From these results, the authors conclude that cFLIP protects ILC1/NK differentiation by protecting differentiating cells from death ligand-induced apoptosis.

This work revealed the importance of STAT5-mediated upregulation of c-FLIP for the differentiation of NKp46+ ILC1/3 and a protective role of ILC1/NK cells in colitis in T/B sufficient mice. The topics are of interest and the results are informative to the readers. There are, however, many concerns as listed below.

- 1) The authors conclude that NKp46+ ILC1 as well as cNK cells are protective for DSS colitis but the mechanisms are largely untouched. Does IFN γ play any role in this protective effect? The authors demonstrated the reduction of IL-12p70, MCP-1 and GM-CSF in c-FLIP<flox/flox>NKp46<iCre> mice but cells producing those cytokines are not shown. Are ILC1/NK cells major producers of those cytokines or other cell types are activated by ILC1/NK cells to produce those cytokines?
- 2) In many inflammatory conditions, CD11b+Gr1+ so-called myeloid suppressor cells are induced and play an anti-inflammatory role. It is informative to examine the induction of such population by ILC1/NK cells.
- 3) The authors used RAG1-/-IL-7R-/- mice to show in Fig. 2 that cNK cells are not affected by the lack of IL-7 signals but all other ILCs are reduced. The authors used in Fig. 1 IL-15-/- and IL-15R-/- mice to show the reduction of cNK cells but they did not perform detailed analysis as Fig. 2. The authors should show the frequency and numbers of NK1.1+ ILC1, T-bet+ ILC1, EOMES+ cNK and ROR γ t+ ILC3 in RAG1-/-IL-15-/- or RAG1-/-IL-15R-/- mice as well.
- 4) The authors examined gut microbiota compositions and discussed a possibility that increase of certain bacterial species such as Turicibacter and Bacteroides could lead to inflammation. However, there is no direct evidence to show that these bacteria play a major role on the inflammation. It is similarly possible that the changes in bacterial species is just a result of inflammation. What happened if WT and c-FLIP<flox/flox>NKp46<iCre> mice were co-housed before and during DSS

chronic colitis induction?

5) The authors showed IFN γ expression by YFP knockin mice in Fig. 7. Are the cells activated or left unstimulated?

Reviewer #2 (ILC differentiation, cytokine)(Remarks to the Author):

The authors investigate the maintenance of NKp46+ ILC subsets in mice lacking the Cflar gene in all NKp46+ cells (Ncr1-Cre; Cflar/Flipfl/fl mice). They observe that splenic NK cells were >90% reduced in these mice and the fraction of NK cells undergoing apoptosis was increased whereas the population of bone marrow NK "precursors" was enriched. In the intestine, all NKp46+ ILC (i.e., cNK, ILC1 and NKp46+ ILC3) were reduced in numbers whereas ILC2 were unaffected. Mice lacking c-FLIP in all NKp46+ cells developed more severe acute and chronic DSS colitis with inflammation-induced changes in microbial communities. While deletion of the IL-7R on all NKp46+ cells had no effect on colitis severity, preferential depletion of cNK cells with anti-asialo-GM1 led to enhanced colitis scores.

This is an interesting manuscript suggesting that NKp46+ ILC may protect against DSS colitis. The data also extends on previous work (e.g., Klose, Cell 2014) showing that all NKp46+ ILC require IL-15 signaling for maintenance and/or survival. The data is of good quality, but at this stage, it is rather descriptive and needs additional decisive data to better link NKp46+ ILC subsets to DSS-induced colitis.

The following specific issues are raised.

1. I do not understand the title of the paper. The authors show that anti-asialo-GM1 depletion mainly affects cNK cells and leads to more severe colitis. Yet, the title claims that ILC1 are protective in DSS colitis. This is misleading and should be corrected. My understanding of the authors' main message is that cNK cells ameliorate DSS colitis.

2. The data support the view that cNK cells protect against DSS colitis. A previous paper (Hall, Mucosal Immunol 2013) has put forward a similar view. In that paper it was demonstrated that this effect is mediated by NKG2A/neutrophil interactions. NKG2A is not expressed by ILC1 so that this would indeed isolate the colitis-protective effect to cNK cells. The authors should discuss these previous findings and provide more decisive data that cNK cells are protective. This reviewer appreciates that there are currently no mouse models available allowing for selective ILC1 deficiency but gain-of-function experiments by adoptive transfer of wildtype cNK cells, ILC1 or NKp46+ ILC3 into DSS-treated Ncr1-Cre; Cflar/Flipfl/fl mice on a Rag-/- background could be performed.

3. The protective role of cNK cells in DSS colitis should be confirmed using mice lacking all cNK cells, i.e., Eomesfl/fl; Ncr1-Cre mice and in mice lacking NKG2A.

4. The authors show that microbial communities change in Ncr1-Cre; Cflar/Flipfl/fl mice treated with DSS. These findings were not further explored. Would transfer of microbiota from Ncr1-Cre; Cflar/Flipfl/fl mice into wildtype mice treated with DSS worsen pathology.

5. The data in Figure 1D indicate that NK cells from mice lacking cFLIP show enhanced apoptosis. Can

the NKp46+ ILC compartment be rescued by crossing in a Bcl2 transgene?

6. The data in Figures 1E/F are not clear. Which pre-gating has been done?

7. The clinical data in Figures 3-6 should be complemented with representative histology of the colon. In addition, colon length is often also supported by photographs of colon/cecum resections from the various mouse strains and treatment groups.

8. It is pivotal to provide absolute cell numbers for cNK cells, ILC1 and NKp46+ and NKp46- ILC3 after anti-asialo-GM1 treatment.

9. The data in Figure 7 would fit better in the context of Figure 1 and 2.

Reply to Reviewer 1

We would like to thank the reviewer for his/her efforts and helpful comments.

1) Reviewer: *"The authors conclude that NKp46+ ILC1 as well as cNK cells are protective for DSS colitis but the mechanisms are largely untouched. Does **IFN γ** play any role in this **protective effect**? (A). The authors demonstrated the reduction of IL-12p70, MCP-1 and GM-CSF in c-FLIP<flox/flox>NKp46 mice but **cells producing those cytokines are not shown**. Are ILC1/NK cells major producer of those cytokines or other cells types are activated by ILC1/NK cells to produce those cytokines? (B)"*

Reply (A): To our knowledge, IFN- γ has disease-promoting rather than protective effects¹⁻³. Accordingly, IFN- γ ^{-/-} mice are protected from DSS-induced colitis⁴. Similarly, colitis severity was reduced in DSS-treated Rag^{-/-}IFN- γ R^{-/-} mice as compared to Rag^{-/-} controls (Bank et al., unpublished results). Together, this indicates that IFN- γ does not protect from intestinal inflammation but rather aggravates disease.

Reply (B): The relative impact of cytokines on disease modulation can change over time⁵. Based on this we hypothesized that persistent production of a particular cytokine correlates positively with its impact on disease modulation. Initially our cytokine assays were performed during the first regeneration phase at d10 (now Fig. 5G; lower panel). To characterize cytokine production at an earlier time point, we repeated our assays at d3 (new set of data; Fig. 5G; upper panel). Only GM-CSF levels were significantly reduced in NK ^{Δ c-FLIP} mice at both time points. On the contrary, IL-12p70 and MCP-1 were lower only at d10. We therefore analyzed GM-CSF production by colonic LPLs from DSS-treated NK ^{Δ c-FLIP} mice and controls. Our results are summarized in Fig. 5H demonstrating that absolute numbers of NKp46⁺, but not NKp46⁻, GM-CSF-producing ILCs were significantly reduced in NK ^{Δ c-FLIP} mice. Hence, lower numbers of GM-CSF⁺ ILCs may explain the reduced regenerative potential of DSS-treated NK ^{Δ c-FLIP} mice. However, we also found GM-CSF production by other cells (e.g. T cells) and therefore cannot determine yet to which extent ILC-derived GM-CSF protects mice from DSS-colitis. For this purpose, NKp46^{iCre} x GM-CSF^{fl/fl} mice would be required. However, their time-consuming, labor-intensive and expensive generation and characterization would be beyond the scope of the current manuscript.

2) Reviewer: *"In many inflammatory conditions, CD11b+Gr1+ so-called myeloid suppressor cells are induced and play anti-inflammatory role. It is of informative to examine the induction of such population by ILC1/NK cells."*

Reply: In the previous version of our manuscript (Fig. 3J/K), we defined **CD11b⁺Gr1⁺** cells as **neutrophils**. However, as pointed out by the reviewer, these markers are

also used to characterize myeloid-derived suppressor cells (MDSCs). Shaul and Fridlender referred to this dilemma in 2017 as follows⁶: “*The emergence of MDSCs, initially defined in mice as CD11b/GR1 double-positive cells, adds an additional complexity to the understanding of the different cancer-driven myeloid subpopulations [37, 38]. MDSCs are immature myeloid cells that do not terminally differentiate into granulocytes, macrophages, or DCs and exhibit immunosuppressive functions by multiple mechanisms. Unfortunately, no clear membrane marker currently exists to differentiate between neutrophils and granulocytic MDSCs in the circulation of tumor-bearing mice or patients with cancer. The GR1 marker includes both the granulocytic Ly6G and the monocytic Ly6C Ags [39, 40]; therefore, discrimination and classification of granulocytic MDSCs vs. TANs remains a subject of debate.*” (TANs = tumor-associated neutrophils)

In addition to their cell surface phenotype, MDSCs and neutrophils share functional features. For example, and contrary to their usual anti-inflammatory function in tumor models, Katoh et al. observed that MDSCs promote DSS-induced colitis⁷ similar to what was described for neutrophils by Hall et al.⁸. Hence, it appears impossible to discriminate between MDSCs and neutrophils in our experimental system. We therefore modified the text accordingly and discuss different possibilities regarding the identity and function of CD11b⁺Gr1⁺ cells in DSS colitis.

3) Reviewer: “*The authors used RAG1^{-/-}/IL-7R^{-/-} mice to show in Fig. 2 that cNK cells are not affected by the lack of IL-7 signals but all other ILCs are reduced. The authors used in Fig. 1 IL-15^{-/-} and IL-15R^{-/-} mice to show the reduction of cNK cells but they did not perform detailed analysis as Fig. 2. The authors should show the frequency and numbers of NK1.1⁺ ILC1, T-bet⁺ ILC1, EOMES⁺ cNK and RORγt⁺ ILC3 in RAG1^{-/-}/IL-15^{-/-} or RAG1^{-/-}/IL-15R^{-/-} mice as well.*”

Reply: To address this point, we analyzed SI LPLs from IL-15^{-/-}, IL-15Rα^{-/-} and WT mice (Suppl. Figure 1). In accordance with Robinette et al.⁹, we observed a strong reduction of cNKs and ILC1s in IL-15^{-/-}/IL-15Rα^{-/-} mice. The abundance of other ILC subsets was not changed significantly. We used mice on a Rag^{+/+} background to make these SI LPL data comparable to those obtained with IL-15^{-/-}/IL-15Rα^{-/-} splenocytes (Fig. 1C-F).

4) Reviewer: “*The authors examined gut microbiota compositions and discussed a possibility that increase of certain bacterial species such as Turicibacter and Bacteroides could lead to inflammation. However, there is no direct evidence to show that these bacteria play major role on the inflammation. It is similarly possible that the changes in bacterial species is just a results of inflammation (A). What happened if WT and c-FLIP<flox/flox>NKp46 mice were co-housed before and during DSS chronic colitis induction?(B)*”

Reply (A): We agree with the reviewer. Based on our results we cannot conclude that alterations of the microbiota drive inflammation. Alternatively, alterations of the

microbiota may be a result of inflammation. In the previous manuscript we therefore stated in the discussion:

*“This may, at least partially, rely on the **disease-related control of the commensal microbiota**. Of note, the lack of NKp46⁺ ILCs in NK^{Δc-FLIP} mice did not have a significant impact on the composition of the commensal microbiota prior to DSS colitis. This argues against a direct impact of NKp46⁺ ILCs on the establishment of the microbiota under homeostatic conditions in T/B-competent mice. Nevertheless, **after establishment of chronic colitis** the composition of the commensal microbiota changed in an NKp46⁺ ILC-dependent fashion.”*

To further emphasize the point raised by the reviewer, we included the following sentences to the discussion:

“Whether the observed alterations of the commensal microflora are a result and/or a driver of inflammation remains an important question for the future.”

Reply (B): For these experiments control littermates were used. Whenever possible, they shared cages from birth on and throughout the experiments. We modified the text accordingly. On page 9 we state:

“To minimize potential cage-specific variations in microbiota composition, NK^{WT} and NK^{Δc-FLIP} mice littermates were, whenever possible, co-housed permanently.”

Furthermore, on page 10 emphasize:

“Notably, genotype-dependent differences in microbiome composition were not visible before induction of DSS colitis (Fig. 6A and B; ADONIS values).”

5) Reviewer: *“The authors showed IFN γ expression by YFP knockin mice in Fig. 7. Are the cells activated or left unstimulated?”*

Reply (B): Cells were isolated from untreated reporter mice, surface molecules were stained and samples were isolated immediately by flow cytometry. To clarify this point we modified the legend of Fig. 2 accordingly:

*“(A) To determine *Ifng* promoter activity in unstimulated ILC precursors, freshly isolated BM cells from eYFP-transgenic IFN- γ reporter mice were analyzed by flow cytometry.”*

References

1. Fuchs, A. ILC1s in Tissue Inflammation and Infection. *Front Immunol* **7**, 104 (2016).
2. Peters, C. P., Mjösberg, J. M., Bernink, J. H. & Spits, H. Innate lymphoid cells in inflammatory bowel diseases. *Immunology Letters* **172**, 124-131 (2016).
3. Neurath, M. F. Targeting immune cell circuits and trafficking in inflammatory bowel disease. *Nat Immunol* (2019).
4. Nava, P. et al. Interferon-gamma regulates intestinal epithelial homeostasis through converging beta-catenin signaling pathways. *Immunity* **32**, 392-402 (2010).
5. Eftychi, C. et al. Temporally Distinct Functions of the Cytokines IL-12 and IL-23 Drive Chronic Colon Inflammation in Response to Intestinal Barrier Impairment. *Immunity* **51**, 367-380.e4 (2019).
6. Shaul, M. E. & Fridlender, Z. G. Neutrophils as active regulators of the immune system in the tumor microenvironment. *J Leukoc Biol* **102**, 343-349 (2017).
7. Kato, H. et al. CXCR2-expressing myeloid-derived suppressor cells are essential to promote colitis-associated tumorigenesis. *Cancer Cell* **24**, 631-644 (2013).
8. Hall, L. J. et al. Natural killer cells protect mice from DSS-induced colitis by regulating neutrophil function via the NKG2A receptor. *Mucosal Immunol* **6**, 1016-1026 (2013).
9. Robinette, M. L. et al. IL-15 sustains IL-7R-independent ILC2 and ILC3 development. *Nat Commun* **8**, 14601 (2017).

Reply to Reviewer 2

We would like to thank the reviewer for his/her efforts and helpful comments.

1) Reviewer: *“I do not understand the **title** of the paper. The authors show that anti-asialo-GM1 depletion mainly affects cNK cells and leads to more severe colitis. Yet, the title claims that ILC1 are protective in DSS colitis. This is misleading and should be **corrected**. My understanding of the authors’ main message is that cNK cells ameliorate DSS colitis.”*

Reply: We agree and changed the title to:

The anti-apoptotic molecule c-FLIP is crucial for the IL-7/IL-15-dependent development of NKp46⁺ ILCs and protection from intestinal inflammation by cNKs

2) Reviewer: *“The data support the view that cNK cells protect against DSS colitis. A previous paper (Hall, Mucosal Immunol 2013) has put forward a similar view. In that paper it was demonstrated that this effect is mediated by NKG2A/neutrophil interactions. **NKG2A is not expressed by ILC1** so that this would indeed isolate the colitis-protective effect to cNK cells. The authors should **discuss these previous findings (A)** and **provide more decisive data that cNK cells are protective (B)**. This reviewer appreciates that there are currently no mouse models available allowing for selective ILC1 deficiency but gain-of-function experiments by adoptive transfer of wildtype cNK cells, ILC1 or NKp46⁺ ILC3 into DSS-treated Ncr1-Cre; Cflar/Flipfl/fl mice on a Rag^{-/-} background could be performed.”*

Reply (A): There is evidence that ILC1s^{1,2} and CD8⁺ T cells^{3,4}, besides cNKs, express NKG2A. Based on this, we have discussed the results of Hall et al.⁵ in more detail and support their interpretation with our findings.

Reply (B): To provide more decisive data on the protective role of cNKs in our disease model, we have generated **NK^{ΔEomes} (NKp46^{iCre} x Eomes^{fl/fl})** mice, which are known to suffer from a selective cNK deficiency⁶⁻⁸. As shown in Fig. 8, NK^{ΔEomes} mice were clearly more susceptible to DSS colitis than controls. Furthermore, numbers of ILC1s were normal and PMN frequencies correlated positively with disease severity, similar to what we had observed in NK^{Δc-FLIP} mice (Fig. 4). Importantly, the overall disease score of **NK^{ΔEomes}** mice (Fig. 8) was comparable to NK^{Δc-FLIP} mice (Fig. 4) strongly suggesting a dominant protective function of cNKs in our model.

3) Reviewer: *“The protective role of cNK cells in DSS colitis should be confirmed using mice lacking all cNK cells, i.e., Eomesfl/fl; Ncr1-Cre mice and in mice lacking NKG2A.”*

Reply: We would like to refer the reviewer to Reply 2B.

4) Reviewer: *“The authors show that microbial communities change in Ncr1-Cre; Cflar/Flipfl/fl mice treated with DSS. These findings were not further explored. Would transfer of microbiota from Ncr1-Cre; Cflar/Flipfl/fl mice into wildtype mice treated with DSS worsen pathology.”*

Reply: Based on our results we cannot conclude that alterations of the microbiota drive inflammation. Alternatively, alterations of the microbiota may be a consequence of inflammation without any disease-promoting effects. To clarify this point, we included the following sentence to the discussion:

“Whether alterations of the commensal microflora are a cause or a result of inflammation remains an important question for the future.”

We hesitated to perform the microbiota transplantation experiments suggested by the reviewer. They are extremely time-consuming, labor- and money-intensive and can be considered as a separate project. This would be far beyond the scope of this manuscript.

5) Reviewer: *“The data in Figure 1D indicate that NK cells from mice lacking cFLIP show enhanced apoptosis. Can the NKp46⁺ ILC compartment be rescued by crossing in a Bcl2 transgene?”*

Reply: Indeed, Bcl-2 has been shown recently to be important for survival of mature NK cells⁹. Using a hypomorphic Bcl-2 allele generated by ENU mutagenesis, the authors could show that reduced Bcl-2 expression leads to preferential loss of mature, i.e. NK1.1⁺ NKp46⁺ CD27⁺ CD11b⁺ and NK1.1⁺ NKp46⁺ CD27⁻ CD11b⁺, cells in the bone marrow. This effect was even greater in spleen. Although the numbers of immature bone marrow NK cells were slightly reduced as well, the frequencies of the different maturation states (as defined by NK1.1 and NKp46 expression) were unaffected. Thus, one can conclude that Bcl-2 affects apoptosis sensitivity at a late stage in life of NK cells. We, however, observed a skewing in NK^{Δc-FLIP} mice towards NKp46 negative populations of NK cell maturation in the bone marrow, which indicates that c-FLIP regulates apoptosis sensitivity early in life of a NK cell.

We also demonstrated by genetic and biochemical means that c-FLIP is downstream of the IL-15/IL-15 receptor/STAT5 pathway. Importantly, it has been previously shown that in γ_c -deficient mice (γ_c being an essential signaling component of the IL-15 receptor) NK cell development cannot be rescued by a Bcl-2 transgene¹⁰. Therefore, we conclude that c-FLIP and Bcl-2 regulate separate survival pathways in NK cells.

This is in line with many reports showing that the extrinsic (i.e. death receptor) apoptosis pathway, which is inhibited by c-FLIP, and the intrinsic (i.e. mitochondrial) apoptosis pathway, which is inhibited by Bcl-2, are independent signaling pathways in most cell types. For instance, a Bcl-2 transgene is not able to rescue activated T cells from CD95/Fas-mediated apoptosis¹¹.

6) Reviewer: “The data in Figures 1E/F are not clear. Which pre-gating has been done?”

Reply: In the corresponding figure legend we state:

“(C-F) After gating on Lin⁻CD122⁺ cells,…”

To specify this in more detail, we provide the reviewer with the following figure depicting our gating strategy:

7) Reviewer: “The clinical data in Figures 3-6 should be complemented with representative histology of the colon. In addition, colon length is often also supported by photographs of colon/cecum resections from the various mouse strains and treatment groups.”

Reply: The requested data were included.

8) Reviewer: “It is pivotal to provide absolute cell numbers for cNK cells, ILC1 and NKp46⁺ and NKp46⁻ ILC3 after anti-asialo-GM1 treatment.”

Reply: The requested data were included (**new** Suppl. Fig. 2).

9) Reviewer: “The data in Figure 7 would fit better in the context of Figure 1 and 2.”

Reply: We agree with this point. Former Figure 7 is now Figure 2.

References

1. Fuchs, A. et al. Intraepithelial type 1 innate lymphoid cells are a unique subset of IL-12- and IL-15-responsive IFN- γ -producing cells. *Immunity* **38**, 769-781 (2013).
2. Krueger, P. D. et al. Murine liver-resident group 1 innate lymphoid cells regulate optimal priming of anti-viral CD8 + T cells. *Journal of Leukocyte Biology* **101**, 329-338 (2017).
3. Rapaport, A. S. et al. The Inhibitory Receptor NKG2A Sustains Virus-Specific CD8⁺ T Cells in Response to a Lethal Poxvirus Infection. *Immunity* **43**, 1112-1124 (2015).
4. André, P. et al. Anti-NKG2A mAb Is a Checkpoint Inhibitor that Promotes Anti-tumor Immunity by Unleashing Both T and NK Cells. *Cell* **175**, 1731-1743.e13 (2018).
5. Hall, L. J. et al. Natural killer cells protect mice from DSS-induced colitis by regulating neutrophil function via the NKG2A receptor. *Mucosal Immunol* **6**, 1016-1026 (2013).
6. Pikovskaya, O. et al. Cutting Edge: Eomesodermin Is Sufficient To Direct Type 1 Innate Lymphocyte Development into the Conventional NK Lineage. *J Immunol* **196**, 1449-1454 (2016).
7. Kwong, B. et al. T-bet-dependent NKp46⁺ innate lymphoid cells regulate the onset of T_H17-induced neuroinflammation. *Nat Immunol* **18**, 1117-1127 (2017).
8. Weizman, O. E. et al. ILC1 Confer Early Host Protection at Initial Sites of Viral Infection. *Cell* **171**, 795-808.e12 (2017).
9. Viant, C. et al. Cell cycle progression dictates the requirement for BCL2 in natural killer cell survival. *J Exp Med* **214**, 491-510 (2017).
10. Kondo, M., Akashi, K., Domen, J., Sugamura, K. & Weissman, I. L. Bcl-2 rescues T lymphopoiesis, but not B or NK cell development, in common gamma chain-deficient mice. *Immunity* **7**, 155-162 (1997).
11. Strasser, A., Harris, A. W., Huang, D. C., Krammer, P. H. & Cory, S. Bcl-2 and Fas/APO-1 regulate distinct pathways to lymphocyte apoptosis. *EMBO J* **14**, 6136-6147 (1995).

REVIEWERS' COMMENTS:

Reviewer #1 (Remarks to the Author):

Bank et al. performed additional experiments and revised their paper. The authors addressed concerns raised by reviewers and the paper is substantially improved. There are, however, still minor points that need to be addressed.

- 1) It is better to cite figures in results section rather than discussion section. It is therefore suggested to move Supplemental Figures 3 and 4 to results section. Supplemental figures.
- 2) Volume and page numbers are missing for references 12, 19, and 63.

Reviewer #2 (Remarks to the Author):

This is a revised manuscript that is significantly improved. The authors have added new and decisive data now linking cNK cells to the protective effect observed in their model system. I support publication of this manuscript.

Response to Referees

Reviewer #1:

Bank et al. performed additional experiments and revised their paper. The authors addressed concerns raised by reviewers and the paper is substantially improved. There are, however, still minor points that need to be addressed.

1) It is better to cite figures in results section rather than discussion section. It is therefore suggested to move Supplemental Figures 3 and 4 to results section. Supplemental figures.

Reply: In the current version of our manuscript, all Supplemental Figures are mentioned in the results section. The order of the Supplemental Figures was changed accordingly.

2) Volume and page numbers are missing for references 12, 19, and 63.

Reply: This was corrected.

Reviewer #2:

This is a revised manuscript that is significantly improved. The authors have added new and decisive data now linking cNK cells to the protective effect observed in their model system. I support publication of this manuscript.